# Stress conditions promote *Leishmania* hybridization in vitro marked by expression of the ancestral gamete fusogen HAP2 as revealed by single-cell RNA-seq

Isabelle Louradour[1†‡], Tiago Rodrigues Ferreira[1†], Emma Duge[1], Nadira Karunaweera[2], Andrea Paun[1], David Sacks[1]*

[1]Laboratory of Parasitic Diseases, National Institute of Allergy and Infectious Diseases, Bethesda, United States; [2]Department of Parasitology, Faculty of Medicine, University of Colombo, Colombo, Sri Lanka

**\*For correspondence:**
dsacks@nih.gov

†these authors contributed equally

**Present address:** ‡Department of Parasites and Insect Vectors, Institut Pasteur/INSERM U1201, Paris, France

**Competing interest:** The authors declare that no competing interests exist.

**Abstract** *Leishmania* are protozoan parasites transmitted by the bite of sand fly vectors producing a wide spectrum of diseases in their mammalian hosts. These diverse clinical outcomes are directly associated with parasite strain and species diversity. Although *Leishmania* reproduction is mainly clonal, a cryptic sexual cycle capable of producing hybrid genotypes has been inferred from population genetic studies and directly demonstrated by laboratory crosses. Experimentally, mating competence has been largely confined to promastigotes developing in the sand fly midgut. The ability to hybridize culture promastigotes in vitro has been limited so far to low-efficiency crosses between two *Leishmania tropica* strains, L747 and MA37, that mate with high efficiency in flies. Here, we show that exposure of promastigote cultures to DNA damage stress produces a remarkably enhanced efficiency of in vitro hybridization of the *L. tropica* strains and extends to other species, including *Leishmania donovani*, *Leishmania infantum*, and *Leishmania braziliensis*, a capacity to generate intra- and interspecific hybrids. Whole-genome sequencing and total DNA content analyses indicate that the hybrids are in each case full genome, mostly tetraploid hybrids. Single-cell RNA sequencing of the L747 and MA37 parental lines highlights the transcriptome heterogeneity of culture promastigotes and reveals discrete clusters that emerge post-irradiation in which genes potentially involved in genetic exchange are expressed, including the ancestral gamete fusogen *HAP2*. By generating reporter constructs for HAP2, we could select for promastigotes that could either hybridize or not in vitro. Overall, this work reveals that there are specific populations involved in *Leishmania* hybridization associated with a discernible transcriptomic signature, and that stress facilitated in vitro hybridization can be a transformative approach to generate large numbers of hybrid genotypes between diverse species and strains.

## Editor's evaluation

In this paper, the authors show that the ability of *Leishmania* promastigotes (a life-cycle stage found in sandflies) to fuse with each other is greatly enhanced after treatments that induce DNA damage. Although the fusion-competent cells express the gamete fusogen HAP2, the parasites do not undergo meiosis and the cells that result from fusion of two diploid organisms are mostly tetraploid; diploid progeny were not recovered. These observations are fundamentally interesting, and could be used for some types of investigations of gene function.

## Introduction

Protozoan parasites of the *Leishmania* genus produce a spectrum of diseases in their mammalian hosts, including humans, ranging from self-limiting cutaneous lesions and tissue-destructive mucosal involvement to infection of the deep viscera that is fatal in the absence of treatment. These clinical outcomes are associated with an extraordinary diversity of *Leishmania* species and strains, with more than 20 species that are pathogenic to humans. All members of the genus have a dimorphic life cycle consisting of extracellular promastigotes that multiply asexually in the digestive tract of their sand fly vectors, and amastigotes that multiply asexually in the phagosomes of their mammalian host cells, primarily macrophages. *Leishmania* were long considered to be essentially clonal, with genetic diversity arising from gradual accumulation of somatic mutations (*Tibayrenc et al., 1993*).

The reproductive strategies of *Leishmania* are now known to include a cryptic sexual cycle that has been inferred from the analysis of hybrid genotypes observed in natural isolates (*Bañuls et al., 1997*; *Dujardin et al., 1995*; *Nolder et al., 2007*; *Odiwuor et al., 2011*; *Ravel et al., 2006*; *Rogers et al., 2014*) and directly demonstrated by the generation of hybrids between different *Leishmania* strains and species in the laboratory (*Akopyants et al., 2009*; *Inbar et al., 2013*; *Romano et al., 2014*; *Sadlova et al., 2011*). The latter involved mating between extracellular promastigotes stages developing in the sand fly and was achieved by coinfection of flies with two parental lines bearing different drug resistance markers, with subsequent selection of double-drug-resistant cells. Experimentally at least, the sexual cycle is nonobligatory, relatively rare, and confined to life-cycle stages present in the sand fly midgut. Based on whole-genome sequencing analyses, the allele inheritance patterns of experimental hybrids provide strong evidence that the system of genetic exchange in *Leishmania* is Mendelian and involves meiosis-like sexual recombination (*Inbar et al., 2019*). Homologues of meiosis-specific genes are found within the genomes of *Leishmania* and are expressed by stages in the sand fly, including the core meiotic genes *SPO11, HOP1,* and *DMC1* involved in creating DNA double-strand breaks, homologous chromosome alignment, and recombination (*Inbar et al., 2017*; *Ramesh et al., 2005*). Homologues of the cell and nuclear fusion proteins HAP2 and GEX1, respectively, have also been found in ancestral trypanosomatid lineages and extant *Leishmania* species, providing indirect evidence that sex was already present in the last eukaryotic common ancestor (*Speijer et al., 2015*). Importantly, there has been no direct evidence that any of these genes or their products are involved in *Leishmania* genetic exchange, and due to the difficulty of making direct observations of the mating events in the sand fly, the precise nature of this reproductive process, including the identification of putative gametic cells, has been difficult to study.

We recently reported that *Leishmania* promastigotes, which can be grown readily in axenic culture, can form stable hybrids entirely in vitro (*Louradour et al., 2020*). While this result proved that mating competent cells can arise in culture, its broader applicability to study the biology of mating and to generate recombinant parasites for genetic linkage analysis was limited by the fact that the in vitro hybridization was confined to only two strains of *Leishmania tropica* and occurred at frequencies far lower than in the sand fly. In the current studies, we have investigated the conditions that might potentiate the mating success of *Leishmania* in vitro. It is common for organisms that are facultatively sexual to undergo sex in response to environmental stress, including conditions that produce DNA damage (*Ram and Hadany, 2016*; *Schoustra et al., 2010*). Thus, a primary function of sex is proposed to be an adaptation for DNA repair, for which a homologous, undamaged chromosome serves as a template (*Bernstein et al., 2018*). Amongst eukaryotic microbes, oxidative DNA-damaging conditions have been shown to promote sexual spore formation in the yeast *Schizosaccharomyces pombe* (*Bernstein and Johns, 1989*) and in the multicellular green alga *Volvox carteri* (*Nedelcu et al., 2004*). Oxidative stress also efficiently induced the expression of sexual pheromone precursors and same-sex mating in *Candida albicans* (*Guan et al., 2019*). DNA damage caused by X-irradiation induced meiotic recombination in the budding yeast *Sacchromyces cerevisiae* (*Thorne and Byers, 1993*) and in the nematode *Caenorhabditis elegans* (*Dernburg et al., 1998*). In the current studies, we have used DNA-damaging conditions to promote in vitro hybridization of *Leishmania*. Exposure to $H_2O_2$, methyl methanesulfonate (MMS), or to γ-radiation produced a remarkably enhanced efficiency in the in vitro hybridization of the *L. tropica* strains, and extended to other species and strains a capacity for in vitro hybridization. Single-cell RNA-seq comparisons of untreated and irradiated promastigotes revealed greatly expanded subpopulations amongst the irradiated cells that expressed homologues of genes involved in genetic exchange. We identified

expression of the ancestral gamete fusogen *HAP2* as a selectable marker for promastigotes able to hybridize in vitro.

## Results

### Stress conditions increase the frequency of *Leishmania* hybrid formation in vitro

We recently developed a protocol for generating *Leishmania* hybrids completely in vitro using two *L. tropica* strains, L747 and MA37, each transfected with different drug resistance and fluorescent markers (*Louradour et al., 2020*). While this protocol achieved the recovery of stable, full-genome hybrids, referred to as LMA hybrids, the frequency of in vitro hybridization was notably low (minimum frequency of hybridizing cells ranging between $10^{-7}$ and $10^{-9}$), particularly in comparison to the frequency of LMA hybrids generated in sand fly coinfection experiments. Stress conditions leading to DNA damage have been shown in other organisms to trigger their sexual reproductive cycles (*Bernstein et al., 2018*; *Ram and Hadany, 2016*; *Schoustra et al., 2010*). Therefore, we chose exposure to γ-radiation, hydrogen peroxide ($H_2O_2$), or MMS, treatments known for their genotoxic effects, as experimental stress conditions. To assess the effects of these treatments on the frequency of hybrid formation in vitro, both the L747 RFP-Hyg and MA37 GFP-Neo parental lines were pretreated or not with the various conditions, and then were mixed without any drug selection and distributed into 96-well plates. For each experiment, the numbers of parental cells present in each well, counted at the moment of mixing, are shown in *Supplementary file 1*. After 24 hr, the content of each well was transferred into double-drug-selective medium (hygromycin and neomycin) in 24-well plates. In untreated conditions, a low proportion (1.7%) of wells yielded double-drug-resistant lines (*Figure 1A–C*). When L747 and MA37 parental lines were pre-exposed to either γ-radiation, $H_2O_2$, or MMS, the frequency of positive wells averaged 63.7, 26.5, or 34.5%, respectively, in the four independent experiments conducted for each treatment. These increased frequencies were observed despite the fact that fewer of the pretreated L747 and MA37 cells were present at the initiation of co-culture due to the negative impact of each of these treatments on parasite growth. Growth curves of L747 and MA37 parasites cultures after exposure to γ-radiation, $H_2O_2$ or MMS are shown (*Figure 1—figure supplement 1*). Area under the curve (AUC) analysis of the culture growth showed a total area ratio of 0.71-fold and 0.44-fold in irradiated L747 and MA37 cultures, respectively, compared to their respective untreated controls. The minimal frequency of hybridizing cells for the two parental lines, calculated using the number of input cells at the start of each co-culture and assuming that each positive well contained only one hybridization event, was between $10^{-8}$ and $10^{-9}$ in untreated conditions (*Figure 1D–F*, *Supplementary file 1*). These frequencies increased between 59- and 347-fold when the cultures were submitted to the different stress conditions.

### Pre-exposure to γ-radiation promotes the generation of intraspecific and interspecific *Leishmania* hybrids

Our previous attempts to generate in vitro hybrids involving *Leishmania* strains other than *L. tropica* L747 and MA37 were unsuccessful (*Louradour et al., 2020*). To evaluate the applicability of the current protocol to different species, we performed a series of crosses using different pairs of parental lines exposed or not to γ-radiation. We attempted several intraspecific crosses: *L. tropica* (*Lt*) MA37 × *Lt*Moro; *Leishmania donovani* (*Ld*) Mongi × *Ld*SL2706; *Leishmania braziliensis* (*Lb*) RicX × *Lb*M1; and *Leishmania major* (*Lmj*) FV1 × *Lmj*LV39 (*Table 1*). Double-drug-resistant lines were successfully generated from all of these parental pairings, with the exception of *Lmj*FV1 and *Lmj*LV39. In addition, cross-species hybrids were recovered from *Lt*MA37 and *Leishmania infantum* (*Li*) LLM320 (*Table 1*). All of the hybrids generated in these crosses were obtained using irradiated parents, with the exception of a single *Lt*MA37 × *Lt*Moro hybrid that was generated using untreated cells. As the parental lines also expressed GFP or RFP fluorescence markers, we used flow cytometry as a rapid way to confirm the hybrid nature of the parasites growing in our assays prior to cloning (*Figure 1—figure supplement 2*). We also compared irradiated *L. braziliensis* crosses in a construct-swap approach. Comparatively, the frequencies of hybrid recovery for both MIR (*Lb*M1-RFPHyg × *Lb*RicX-GFPNeo) and RIM (*Lb*RicX-RFPHyg × *Lb*M1-GFPNeo) crosses were identical (*Table 1*), suggesting the reproducibility of

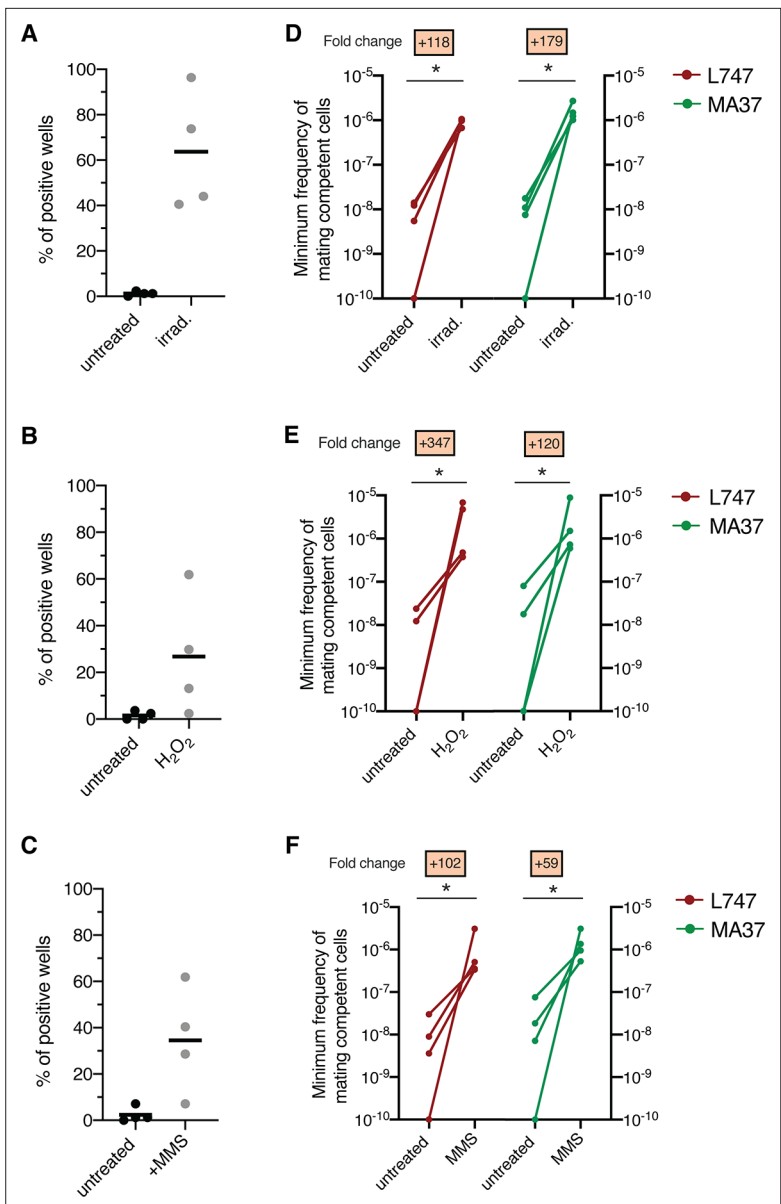

**Figure 1.** Stress treatments strongly enhance in vitro hybridization of *L. tropica* L747 and MA37. (**A–C**) Proportion of culture wells exhibiting growth of LMA hybrids when exposed to γ-radiation (**A**), hydrogen peroxide ($H_2O_2$) (**B**), or methyl methanesulfonate (MMS) (**C**); data are presented as mean values (black line) and individual measurements. (**D, E**) Minimum frequency of hybridization – competent cells for both L747 and MA37 parental strains after exposure to γ-radiation (**D**), $H_2O_2$ (**E**) or MMS (**F**). Four independent experiments were performed for each treatment. The frequencies calculated for the control and treated conditions in each experiment are linked by a line. *p=0.0286 (Mann–Whitney test).

The online version of this article includes the following source data and figure supplement(s) for figure 1:

**Source data 1.** Percentage of hybrid-positive wells in *L. tropica* in vitro crosses using parental lines submitted to different DNA stress.

**Figure supplement 1.** Effect of different stress treatments on *L. tropica* promastigote growth in vitro.

**Figure supplement 2.** Flow cytometry data of a representative double-fluorescent irradiation-facilitated hybrid from the different crosses where each parent expresses either GFP or RFP.

**Table 1.** List of the different intra- and interspecific irradiation-facilitated hybrids (from in vitro hybridization after exposure of parental strains to $\gamma$-radiation) generated in this study with their respective ploidies according to propidium iodide staining, flow cytometry, and whole-genome sequencing.

| | Parental strains | | Hybrid denomination | Frequency of hybrid recovery: # positive wells/# total wells | | Ploidy of irradiation-facilitated hybrids: # hybrids/# hybrids tested | | |
| Type of cross | A | B | | No treatment | Irradiation | 2n | 3n | 4n |
|---|---|---|---|---|---|---|---|---|
| Intraspecific | *L. tropica* L747 RFP-Hyg | *L. tropica* MA37 GFP-Neo | LMA | 4/336 | 214/336 | 0/44 | 4/44 | 40/44 |
| Intraspecific | *L. tropica* Moro RFP-Hyg | *L. tropica* MA37 GFP-Neo | MoMA | 1/84 | 20/84 | 0/4 | 1/4 | 3/4 |
| Interspecific | *L. infantum* LLM320 RFP-Hyg | *L. tropica* MA37 GFP-Neo | IMA | 0/84 | 3/84 | 0/3 | 0/3 | 3/3 |
| Intraspecific | *L. donovani* Mongi RFP-Hyg | *L. donovani* SL2706 GFP-Neo | Mondo06 | 0/84 | 35/84 | 0/16 | 0/16 | 16/16 |
| Intraspecific | *L. braziliensis* M1 RFP-Hyg | *L. braziliensis* RicX GFP-Neo | MIR | 0/84 | 9/84 | 0/7 | 1/7 | 6/7 |
| Intraspecific | *L. braziliensis* RicX RFP-Hyg | *L. braziliensis* M1 GFP-Neo | RIM | 0/84 | 9/84 | 0/6 | 0/6 | 6/6 |
| Intraspecific | *L. major* FV1 Sat | *L. major* LV39 RFP-Hyg | LVFV | 0/84 | 0/84 | * | * | * |

*Not detected.

RFP: red fluorescent protein; GFP: green fluorescent protein; mNG-yTub: mNeonGreen-gammaTubulin fusion; Hyg: hygromycin B resistance marker; Neo: neomycin resistance marker; Bsd: blasticidin S resistance marker.

this protocol independently of the pairwise combination of the two drug resistance markers in the two parental strains.

## Genome ploidy and whole-genome sequencing analyses reveal that in vitro hybrids are full-genome, polyploid hybrids

The double-drug-resistant hybrid lines were cloned by distribution in 96-well plates in promastigote growth medium containing both antibiotics. The DNA content of selected irradiation-facilitated, in vitro hybrid clones was analyzed using propidium iodide (PI) staining followed by flow cytometry. The diploid parental lines were used as controls for normalization. With the exception of four triploid LMA hybrids, one triploid *Lt*MA37 × *Lt*Moro (MoMA) hybrid, and one triploid *L. braziliensis* hybrid, all of the other intra- and interspecific hybrids were close to tetraploid (***Figure 2—figure supplement 1***, ***Table 1***). No diploid hybrids were observed, which distinguishes them from the LMA hybrids recovered from the in vitro LMA hybrids generated from untreated parental lines, for which diploid and polyploid hybrids were obtained (***Louradour et al., 2020***). Confocal microscopy imaging of Hoechst 33342 stained promastigotes and transmission electron microscopy show that the irradiation-facilitated hybrids carry a single nucleus and are not heterokaryons (***Figure 2C and D***).

While polyploid hybrids, including tetraploid progeny, have been recovered from experimental hybrid lines generated in sand flies, the majority of in vivo hybrids have been close to diploid (***Akopyants et al., 2009***; ***Inbar et al., 2013***; ***Inbar et al., 2019***). These results suggest that tetraploid intermediates, as have been described in some fungi, might reflect a normal part of the sexual cycle in *Leishmania*. In this context, under the conditions of irradiation-facilitated hybridization in vitro, promastigotes capable of cell and nuclear fusion can be generated but somehow fail to complete the ploidy reduction steps of the meiotic cycle. In attempts to promote these processes, selected LMA tetraploid clones were re-exposed to stress conditions, including $H_2O_2$ and γ-radiation, and/or were passed through sand flies. All of these attempts to induce ploidy changes in the LMA hybrids were unsuccessful (***Figure 2—figure supplement 2***).

To determine the copy number of individual chromosomes and their respective parental genome contributions, we performed whole-genome sequencing of selected progeny clones. For somy quantification, the average of coverage for each chromosome was scaled to the ploidy of the cells. The somy profiles of the parents and hybrid clones are depicted as heatmaps in ***Figure 2A***. The parental somies are in each case close to 2n, with the exception of chr 31, which is trisomic in every parent,

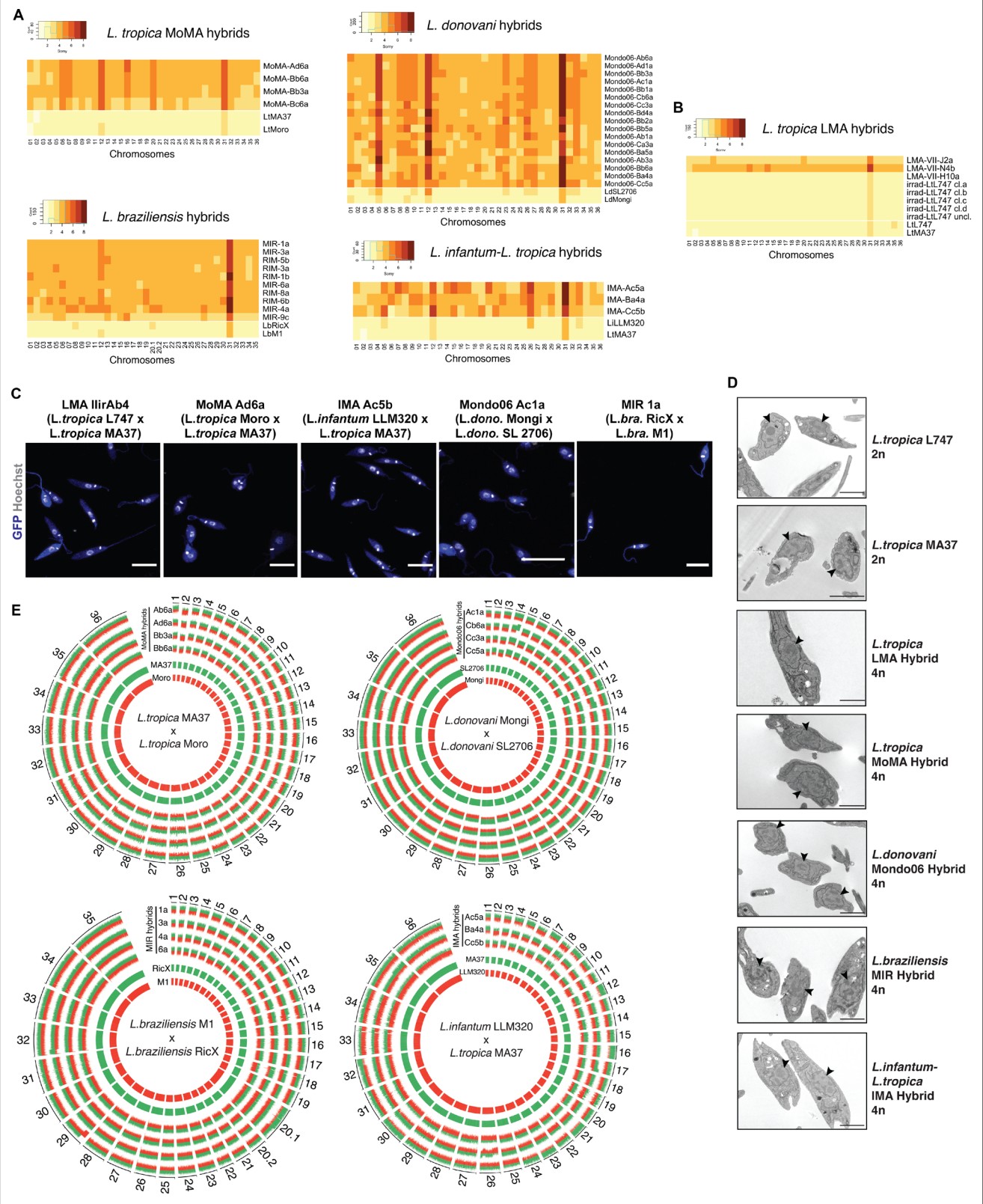

**Figure 2.** Whole-genome sequencing of irradiation-facilitated experimental hybrids reveals chromosome biparental inheritance and recombination breakpoints. (**A**) Normalized sequence read depth of the different *Leishmania* chromosomes (35 chrs in *L. braziliensis* and 36 in the other species shown) was used to infer somy values in parental strains (bottom two rows in each heatmap) and hybrids. (**B**) Normalized sequence read depth of *L. tropica* LMA hybrids of different ploidies generated in untreated in vitro hybridization, and of four independent subclones and one uncloned line of the irradiated

*Figure 2 continued on next page*

*Figure 2 continued*

L747 parent. (**C**) Fluorescence confocal microscopy images of tetraploid LMA, MoMA, IMA, Mondo06, and MIR irradiation-facilitated hybrids. Nuclear and mitochondrial DNA (kDNA) are shown in white (Hoechst 33342 staining) and cell bodies in blue (GFP expression). Scale bar: 10 µm. (**D**) Transmission electron micrographs of representative diploid (2n) parental strains and different tetraploid (4n) hybrids depicting a single nucleus (arrowheads) in promastigote forms not undergoing cell division. Scale bar: 2 µm. (**E**) Circos plots representing the inheritance patterns of all the homozygous SNP differences between the two parental strains in four *L. tropica* MoMA intraspecific hybrids, three *L. infantum-L. tropica* IMA interspecific hybrids, four *L. donovani* Mondo06 intraspecific hybrids, and in four *L. braziliensis* MIR intraspecific hybrids. Each group separated by radial white lines represents a different chromosome, and chromosome ids are shown on the outer circle. Hybrid clones ids are indicated at the start of each circular track. Red and green histograms depict inferred parental contribution from homozygous SNPs specific to the RFP-expressing parental line and the GFP-expressing parental line, respectively. Whole-genome sequencing analyses were performed using the PAINT software (*Shaik et al., 2021*) and reference genomes available on Tritrypdb (tritrypdb.org).

The online version of this article includes the following figure supplement(s) for figure 2:

**Figure supplement 1.** DNA content analysis of irradiation-facilitated hybrids by propidium iodide (PI) staining followed by flow cytometry.

**Figure supplement 2.** Description of the different approaches attempted to reduce the ploidy of LMA irradiation-facilitated hybrids (hybrids LMIIirAb4 and IIirBb1).

**Figure supplement 3.** Circos plots showing inferred parental SNP contribution in all *L. braziliensis and L. donovani* sequenced hybrid genomes.

**Figure supplement 4.** Parental SNP contribution in the kDNA maxicircle sequences from *L. tropica* MoMA hybrids.

along with other chromosomes that show close to 3n profiles in combinations that are specific to the parental line, for example, chrs 8, 20.1 in *Lb*RicX; chrs 5, 12, 26, 33 in *Li*LLM320. The majority of the chromosomes that are 2n in both parents are close to 4n in the hybrid progeny. For those chromosomes that are close to 3n in one of the parents, the hybrid somies are in most cases close to 5n, while chromosomes that are close to 3n in both parents are close to 6n in the hybrids. These inheritance patterns reinforce the conclusion that the hybrids are products of the fusion of the complete parental genomes. There are, nonetheless, a large number of hybrid chromosomes that deviate from the expected additive copy number, showing an apparent gain or loss of somy. As a result, within a particular cross each of the hybrids appear to possess a unique combination of somies. This plasticity was related to hybridization and was not driven by the irradiation per se since four independent subclones and one uncloned line of the irradiated L747 parent did not show any changes in somy from the untreated line (*Figure 2B*).

For all intra- and interspecific hybrids analyzed, we could observe biallelic inheritance of almost all of the nuclear homozygous SNPs that are different between the two parental lines (*Figure 2E*), indicating that they are full-genome hybrids. The SNP inheritance profiles are depicted as Circos plots showing the relative parental contributions of homozygous SNPs across all 35 (chr 20 in *L. braziliensis* is a single chromosome product of a fusion between chrs 20 and 34 but is analyzed as chrs 20.1 and 20.2 in the plot) or 36 chromosomes. While the tetraploid hybrids show generally balanced contributions of chromosomes from both parents, in many cases asymmetric contributions are observed, for example, chrs 12, 16, 21, 30 in IMA (*Lt*MA37 × *Li*LLM320) hybrid Ac5a, for which three copies are contributed by the *Lt*MA37 parent, and one copy by *Li*LLM320. Rare instances of uniparental inheritance involving a particular chromosome are also observed, for example, chr 12 in Mondo06 (*Ld*Mongi × *Ld*SL2706) hybrid Cc5a, for which loss of heterozygosity is inferred. There are a substantial number of chromosomes for which the asymmetric or uniparental inheritance patterns involve only part of the chromosome, for example, chr 32 in IMA Ac5a; chrs 17 and 26 in IMA Ba4a; chrs 4, 13, 34 in Mondo06 Cb6a; chrs 20, 29 in MIR (*Lb*RicX × *Lb*M1) 1a; and chrs 14, 32 in MIR 3a, implying recombination between homologous chromosomes. The asymmetric parental contributions, both partial and complete at the level of individual somy, are highlighted in the Circos plots shown in *Figure 2—figure supplement 3*. For the 17 *Ld*Mongi × *Ld*SL2706 hybrids analyzed, 102 chromosomes, or 17% of the total, show evidence of either recombination or the gain or loss of somy producing unbalanced contribution of the parental alleles.

Additional analysis of SNPs detected in the kDNA (mitochondrial DNA equivalent) of *L. tropica* MoMA crosses indicates the uniparental inheritance of parental maxicircle sequences in the irradiation-facilitated hybrids (*Figure 2—figure supplement 4*). This uniparental maxicircle contribution is consistent with the vast majority, if not all, of the hybridization events from in vivo and natural *Leishmania* crosses described to date (*Akopyants et al., 2009*; *Franssen et al., 2020*; *Inbar et al., 2013*; *Van den Broeck et al., 2020*).

# Single-cell RNA sequencing reveals a unique transcriptomic landscape in parasites exposed to γ-radiation

Although expression of protein-coding genes in kinetoplastids is not regulated at the level of transcription initiation, transcriptomic analyses of kinetoplast populations, including *Leishmania*, have revealed clear differences in transcript levels comparing parasites from different life-cycle stages or from parasites submitted to stress conditions. These levels are controlled by gene dosage and by a post-transcriptional regulatory network involving different RNA-binding proteins (*De Pablos et al., 2016*; *Iantorno et al., 2017*). We used single-cell RNA sequencing (scRNA-seq) between irradiated and nonirradiated promastigotes to help define transcripts and possible rare cell types associated with hybridization in *Leishmania*. Using the Chromium Single Cell 3′ workflow (10X Genomics) and Illumina sequencing, gene expression analysis with the Seurat R toolkit (*Hao et al., 2021*) was carried out on 20,122 and 16,708 cells obtained from duplicate cultures of L747 and MA37, respectively, exposed or not to γ-radiation, 1 day post-inoculation of a log-growth culture at $5 \times 10^6$ promastigotes per mL. After filtering the data to remove potential cell multiplets, dying cells or cells with poor quality transcripts (see Materials and methods), 20,087 L747 (13,392 untreated plus 6695 irradiated) and 16,645 MA37 (8156 and 8489 irradiated) cells remained, with expression of 8156 different genes detected and a median number of 1079 different mRNAs detected per cell. Since the approach used corresponds to a 3′Tag-based sequencing and there are no data available on *L. tropica* transcripts 3′UTR length, we chose *L. major* as the reference transcriptome as it has the highest-quality *Leishmania* gene annotations (*Dillon et al., 2015*) and the genomes of both species are highly similar and syntenic.

To estimate the degree of similarity of the global mRNA expression profiles between the different samples and to evaluate the reproducibility of the replicate samples, we performed a principal component analysis (PCA) using a 'pseudo-bulk' quality control method (DESeq2) (*Love et al., 2014*). The PCA plot shows the top two principal components that explain most of the variance between the samples, 79% and 16% for PCA1 and PCA2, respectively, and high reproducibility between the replicate samples (*Figure 3A*). The analysis suggests that L747 and MA37 cells present distinct mRNA expression profiles, and that the L747 transcriptome is altered more by the irradiation than MA37.

Data from the replicate cultures were combined and analyzed with Seurat R toolkit (*Hao et al., 2021*). The 3000 most variable genes were selected (*Figure 3—figure supplement 1A*) and used for unsupervised clustering. Uniform Manifold Approximation and Projection (UMAP) was used to visualize the relationship between individual transcriptomes in two dimensionality, where variation between transcriptomes dictates the distance between cells. First, analysis of the untreated parasite samples separately from irradiated cells revealed a transcriptomic heterogeneity among promastigotes within the same mid-logarithmic-phase culture, which would not be detectable using bulk-RNA sequencing without prior cell sorting. Clustering of untreated cells identified five and six different cell populations in L747 and MA37, respectively (*Figure 3—figure supplement 1B*), using reduction dimensions 1–7 and a resolution of 0.3. Transcript markers for procyclic (*DOT1B*, *Inbar et al., 2017*) and metacyclic promastigote stages (*SHERP*, *Inbar et al., 2013*), stress-induced messenger ribonucleoprotein (mRNP) granules (*DHH1*, *Kramer et al., 2010*), and protein synthesis (*RPS11*) are among the cluster-specific genes found in untreated samples (*Figure 3—figure supplement 1C–F*). As expected for *Leishmania* cell cultures at early-log growth, the majority of the single-cell gene transcriptomes reflect the predominance of proliferative procyclic promastigotes over other life-cycle stages. The main core of clusters identified in the untreated samples of both strains revealed transcripts found to be upregulated in procyclics by bulk RNA analyses, while cells that upregulate a restricted number of metacyclic-specific genes comprise a small, discrete cluster only in untreated MA37 (cluster 5 in *Figure 3—figure supplement 1B and C*). By contrast, cells expressing *DHH1* were concentrated in a discrete cluster only in L747 (cluster 4 in *Figure 3—figure supplement 1E*).

Integration of the single-cell data of the untreated samples to identify shared cell populations further highlighted differences between the two strains (*Figure 3—figure supplement 2*, *Supplementary file 2*). The standard Seurat pipeline for scRNA-seq data analysis revealed no significant co-clustering of untreated cells. Interestingly, cells with upregulated expression of several RNA-binding proteins involved in RNA metabolism, including *DHH1*, *UBP1*, *UBP2*, *DDX3*, *PABP2*, and *ZC3HC31*, are enriched in a discrete cluster found only in L747 (*Supplementary file 2*, cluster 6). After identification of match cell populations ('anchors') between the strains and integrating the data, new clustering was performed, which again revealed limited overlap between the strains (*Figure 3—figure*

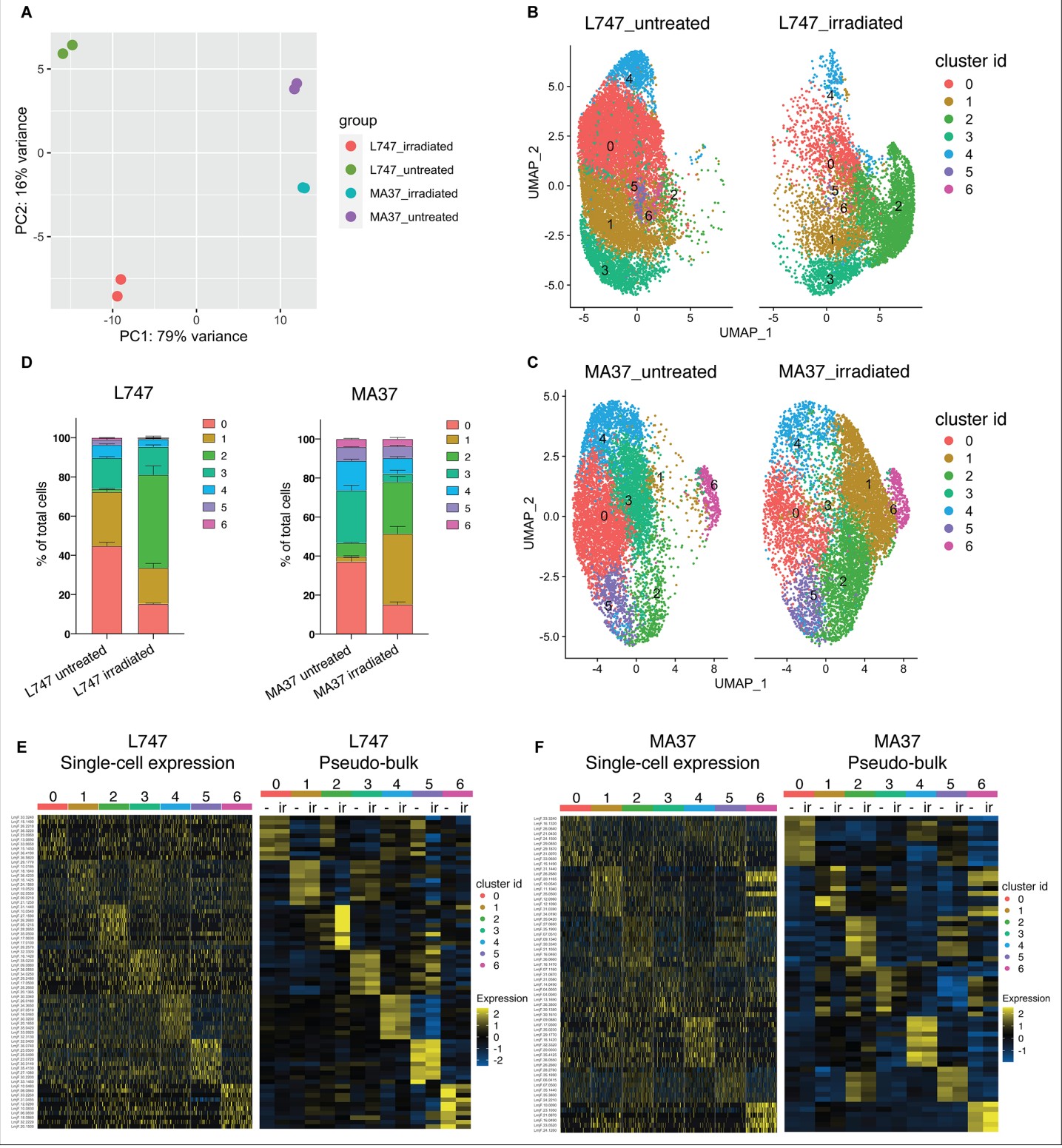

**Figure 3.** Single-cell RNA sequencing analysis of *L. tropica* parental strains identify discrete clusters of cells expanded after irradiation. (**A**) Principal component analysis (PCA) plot of average gene expression comparing scRNA-seq biological replicates of L747 and MA37 promastigote cultures 1 day post-exposure to 6.5 Gy of γ-radiation or not treated (n = 2 for each group). (**B, C**) Unifold Manifold Approximation and Projection (UMAP) visualization of the seven heterogeneous clusters of cells identified in L747 (**B**) and MA37 (**C**) according to their transcriptomic profiles using the Seurat R package (*Hao et al., 2021*). Cluster identities in L747 and MA37 are completely independent and unrelated with each other. Data presented are a combination of two biological scRNA-seq replicates for each group. (**D**) Representation (%) of each cell cluster per sample in irradiated or untreated L747 and MA37

*Figure 3 continued*

in the two replicates. (**E, F**) Left panel: heatmap representation of the expression (log2 of fold change comparing a single cell versus all other cells within the same sample) of the top 10 transcript markers in each cluster identified for L747 (**E**) and MA37 (**F**), downsampled to 100 cells in each cluster for visualization (Wilcoxon rank-sum test adj. p<0.05). Right panel: a pseudo-bulk analysis shows the average gene expression of the same top 10 markers, highlighting the differences and similarities between untreated (-) and irradiated (Ir) samples using data from all cells assigned to each cluster.

The online version of this article includes the following figure supplement(s) for figure 3:

**Figure supplement 1.** Transcriptomic heterogeneity in untreated *L. tropica* strains.

**Figure supplement 2.** Differential single-cell gene expression in integrated scRNA-seq data from untreated L747 and MA37 cells.

*supplement 2*, *Supplementary file 2*). Clusters that are significantly shared between the two strains are marked by the expression of genes encoding ribosomal proteins, cyclins, chaperones, kinesins, and flagellar proteins, which most likely represent a subgroup of proliferative promastigotes (integrated untreated clusters 2 and 4 in *Figure 3—figure supplement 2*, *Supplementary file 2*).

Given the substantial strain differences in the transcriptomes of the untreated cells, the scRNA-seq untreated vs. irradiated data comparisons were performed separately for each strain to avoid potential confounding factors in the cell clustering. We analyzed irradiated and untreated cells together, separating the two strains. Seurat unsupervised clustering with dimensions of reduction 1–7 and a resolution of 0.34 identified seven groups of cells for each strain (*Figure 3B and C*). For both L747 and MA37, submitting the cells to γ-radiation altered the relative distribution of cells within particular clusters, with greatly increased cell frequencies in cluster 2 for the irradiated vs. untreated L747 cells (47.5% vs. 1.5%) and cluster 1 for MA37 (36.2% vs. 2.7%) cells (*Figure 3D*). By contrast, the relative abundance of cells in MA37 cluster 3 decreased after treatment, suggesting that cells from this cluster might be closely related to the irradiation-induced MA37 cluster 1. The dominant cluster 2 in the irradiated L747 culture stands out as a spatially separated cluster, consistent with the PCA plot showing the relatively strong effect that the γ-radiation has on altering the transcriptome of this strain.

Relative expression of the top 10 upregulated transcripts in each of the seven clusters in each strain revealed the different gene expression profiles in the subgroups of cultured promastigotes (*Figure 3E and F*, *Supplementary file 3*). Despite the fact that the UMAP visualizations failed for the most part to resolve the populations into well-separated clusters, the heatmaps showing the different gene expression levels still reveal significant transcriptomic heterogeneity between each of the groups (*Figure 3E and F*). A pseudo-bulk analysis of the scRNA-seq data shows differential expression of the top 10 markers between irradiated and untreated cells only within L747 cluster 2, and within MA37 clusters 1 and 3 (*Figure 3E and F*). Other clusters remain largely unchanged upon irradiation.

The observation that cluster 2 in L747 (L-cluster2) and cluster 1 in MA37 (M-cluster1) are largely expanded upon irradiation prompted us to investigate a possible transcriptomic signature shared between these two promastigote populations when compared with other cells in the same culture. Gene Ontology (GO) analysis revealed that molecular functions such as mRNA binding and kinase activity are significantly enriched in both L-cluster2 and M-cluster1 among the lists of differentially expressed genes (DEGs; |log2FC| > 0.1, FDR < 0.05) (*Figure 4A*). From the 708 genes upregulated post-irradiation in L-cluster2 and 296 genes in M-cluster1, 169 are common between the two strains (*Figure 4B*, *Supplementary file 4*). This list includes repressor of differentiation kinase 2 (*RDK2*; LmjF.31.2960) and RNA-binding protein 5 (*RBP5*; LmjF.09.0060), both of which are involved in *Trypanosoma brucei* proliferation (*Gilabert Carbajo et al., 2021*; *Jones et al., 2014*). Among the top 20 most upregulated genes in either L-cluster2 or M-cluster1 are seven genes common to both lists: serine/threonine-protein kinase *NEK15* (LmjF.26.2570), adenylate-cyclase *ACP2* (LmjF.10.0540), three putative surface antigen proteins (LmjF.05.1215; LmjF.12.1090 and LmjF.35.0550), one gene of unknown function (LmjF.26.2680), and a putative leucine-rich repeat (LRR) protein annotated as a pseudogene on TriTrypDB (LmjF.31.1440) (*Figure 4B*).

Concerning the latter gene, potential orthologs in other *Leishmania* species reference genomes are annotated as hypothetical proteins and there is evidence of a possible multigene family (e.g., in *Leishmania mexicana* LmxM.30.1440 and LmxM.30.1442; in *L. infantum* LINF_310021500 and LINF_310021400), suggesting annotation inconsistency. Among the top 20 most downregulated genes in either cluster that are also downregulated in the other cluster are four genes: two ribosomal proteins (LmjF.04.0950 and LmjF.26.1640), a P-type ATPase (LmjF.18.1510), and translation elongation factor 2 (LmjF.36.0190).

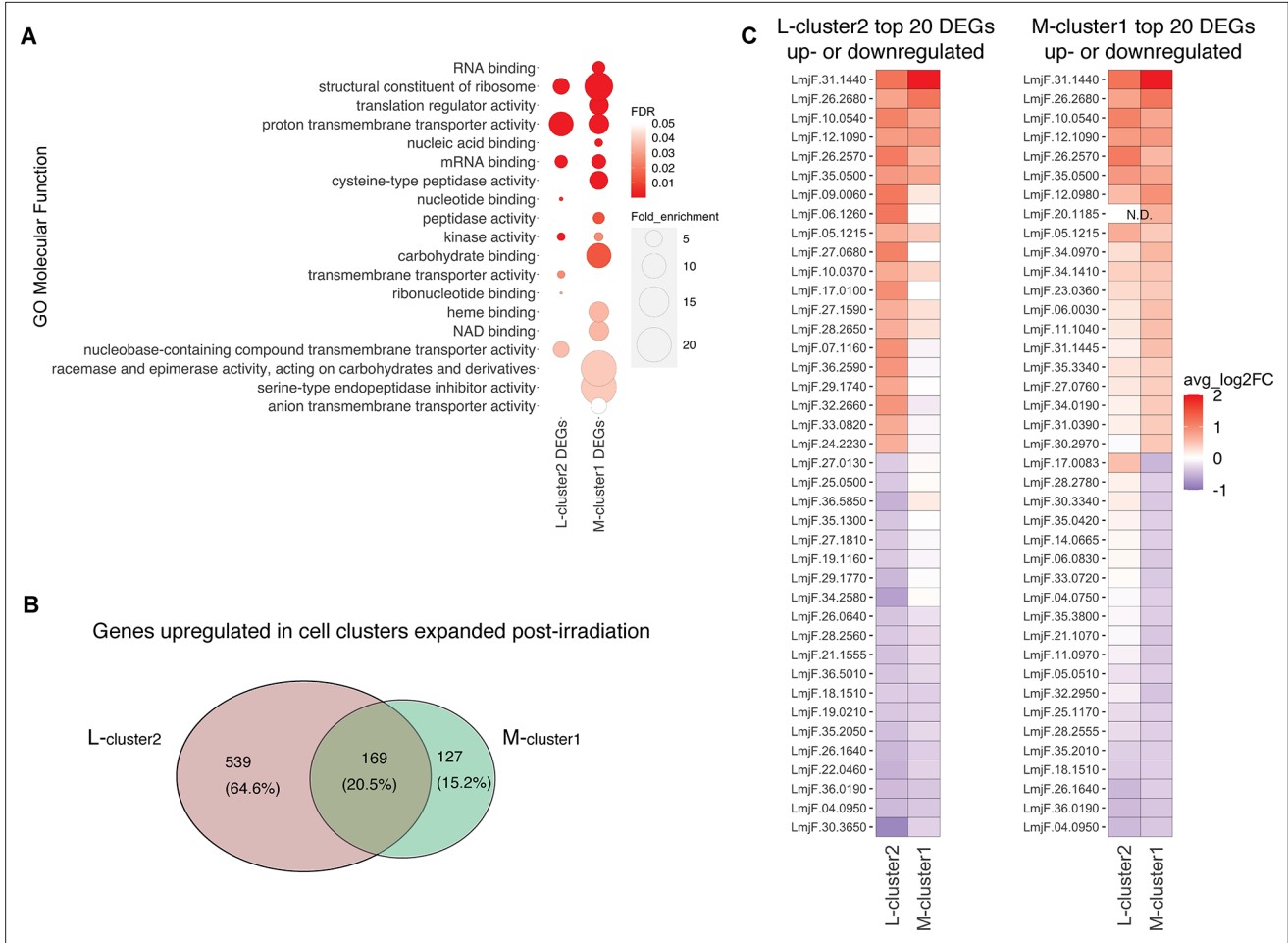

**Figure 4.** Irradiation-induced cell clusters in L747 and MA37 share a common transcriptomic signature. (**A**) Gene Ontology (GO) molecular function enrichment analysis was performed for the lists of differentially expressed genes (DEGs; |log2FC| > 0.1; Wilcoxon rank-sum test adj. p<0.05) in L747 cell cluster 2 (L-cluster2) and MA37 cell cluster 1 (M-cluster1) using TritrypDB online tools (Benjamini–Hochberg FDR < 0.05). (**B**) Venn diagram representation of the total number of genes upregulated in L-cluster2 and/or M-cluster1 compared to the other clusters (log2FC > 0.1; Wilcoxon rank-sum test adj. p<0.05). (**C**) Heatmap representation of the expression of the top 20 genes up- or downregulated in L-cluster2 (left panel) or M-cluster1 (right panel). Data are presented as the average gene expression in cells from each of the clusters compared with the other clusters in the same sample. Genes are ranked according to their mean expression in both clusters. Expression of gene LmjF.20.1185 was not detected (N.D.) in the L-cluster2 cells.

## Homologues of meiotic genes are enriched in unique clusters of irradiated L747 and MA37 cells

We looked at the expression of genes expressed by the irradiated cells that are potentially involved in genetic exchange. From the list of 169 upregulated genes shared between L-cluster2 and M-cluster1, we identified three putative homologues of genes involved in genetic exchange: a gene encoding the ancestral gamete fusogen HAP2/GCS1 (hapless 2/generative cell-specific 1; LmjF.35.0460), the nuclear membrane protein GEX1 (gamete expressed 1; LmjF.04.0100) required for nuclear fusion during yeast mating, and RAD51 recombinase (LmjF.28.0550), involved in DNA-damage repair and genetic exchange in *Trypanosoma cruzi* (*Alves et al., 2018*). The average expression of these and additional meiotic gene homologues was also tested by comparing irradiated and untreated cells within each parental line. HAP2, GEX1, and RAD51 are specifically upregulated in the irradiated L747 and MA37 cells, their expression being detected in 28.4 and 12% of the treated cells, respectively, compared to 11.9 and 5.75% of the untreated cells (*Figure 5A and B*). The single-cell relative expression of HAP2 and GEX1, depicted by UMAP and violin plots, is significantly enriched in L-cluster2 and M-cluster1 in irradiated samples (*Figure 5C–F*) compared to the other cell clusters in any condition.

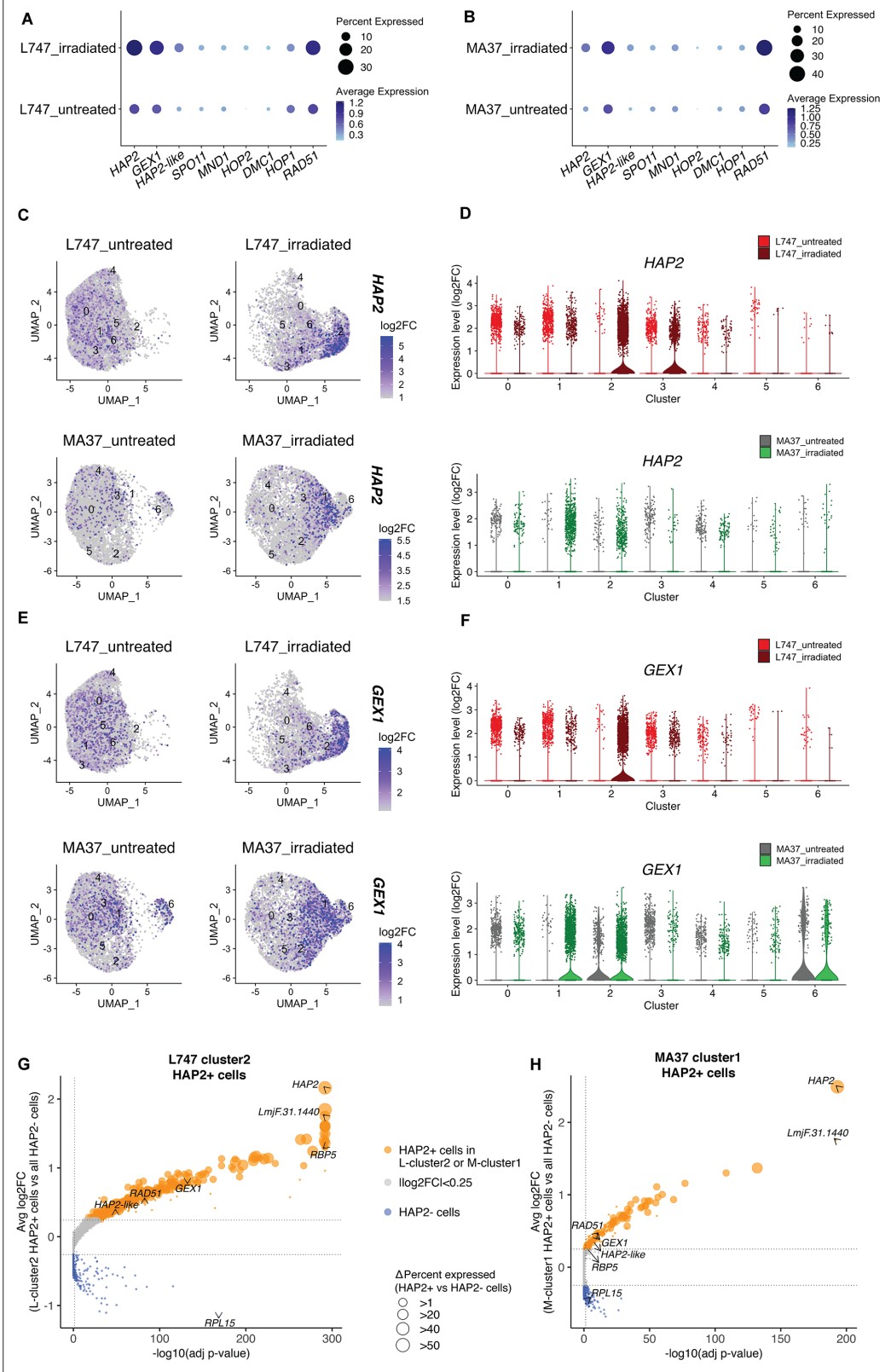

**Figure 5.** Expression of HAP2 and GEX1 is enriched in irradiation-induced L747 cluster 2 and MA37 cluster 1. (**A, B**) Dot plots depicting the average expression of *Leishmania* homologues of genes involved in genetic exchange and meiosis in other organisms, shown for L747 (**A**) and MA37 (**B**) populations with and without exposure to irradiation (pseudo-bulk analysis). The dot size in the plots represents the percentage of cells expressing

*Figure 5 continued on next page*

*Figure 5 continued*

those genes in each sample. Meiotic genes tested are *HAP2* (LmjF.35.0460), *GEX1* (LmjF.04.0100), *HAP2-like* (LmjF.36.3860), *SPO11* (LmjF.16.0630), *MND1* (LmjF.24.1010), *HOP2* (LmjF.27.2420), *DMC1* (LmjF.35.4890), *HOP1* (LmjF.36.1110), and *RAD51* (LmjF.28.0550). (C–E) Unifold Manifold Approximation and Projection (UMAP) plots showing the expression of HAP2 (C) and GEX1 (E) in the different cell clusters identified in L747 (upper panels) and MA37 (lower panels) untreated or irradiated. (D–F) Violin plots representation of the single-cell expression data of HAP2 (D) and Gex1 (F) in L747 (upper panel) and MA37 (lower panel), comparing cells from untreated and irradiated cultures. (G, H) Volcano plot representation of differentially expressed genes (DEGs) in HAP2$^+$ cells (i.e., cells with HAP2 expression log2FC > 1.0) identified in L-cluster2 (G) and M-cluster1 (H). Delta percent expressed (%HAP2$^+$ minus %HAP2$^-$ cells expressing a particular gene) is presented as different data point sizes in the plots. Threshold values for adjusted p<0.05 (vertical dotted line; Wilcoxon rank-sum test) and |log2FC| > 0.25 (horizontal dotted lines) were used.

## HAP2 expression is associated with *L. tropica* hybridization competence in vitro

The well-conserved class II fusion protein HAP2 is involved in the fusion of gametes or opposite mating-type cells in many organisms (*Fédry et al., 2017*). Expression of HAP2 has been previously used to identify gametes and immediate precursors in the related trypanosomatid *T. brucei* found inside the salivary glands of the tsetse fly vector (Peacock et al.). We hypothesized that *L. tropica* HAP2 might be a marker for hybridization-competent cells. First, we investigated the list of genes co-expressed with *HAP2* in each of the irradiation-induced clusters, M-cluster1 and L-cluster2 (gene expression in *HAP2*$^+$ + in each of these clusters vs. the total *HAP2*$^-$ cell population in each strain; |log2FC| > 0.25, adj. p<0.05). Several genes that were upregulated in cells from L-cluster2 and M-cluster1 as a whole (*Figure 4C*), for example, *RBP5* (LmjF.09.0060) and LmjF.31.1440, were also upregulated in the *HAP2*$^+$ cells within these clusters, as were other meiosis-related genes; *GEX1* (LmjF.04.0100), *RAD51* (LmjF.28.0550), and LmjF.36.3860, encoding a HAP2-like protein.

In order to follow the expression of HAP2 at the protein level, we generated reporters of expression by tagging the endogenous protein with a fluorescent mNeonGreen (mNG) fusion in both L747 (*Figure 6A*) and MA37 (*Figure 6F*). In each case, the tag was inserted by CRISPR/Cas9 in the N-terminal part of the ORF, so that the endogenous sequences, in particular the 3′UTR involved in post-transcriptional regulation, would not be affected. The intensity of mNG expression detected by flow cytometry was low in both parental strains, but in both cases a proportion of cells reproducibly expressed a fluorescence level higher than the negative control (L747 or MA37 cultures without the reporter construct; *Figure 6B and G*). Following inoculation of fresh cultures with stationary phase L747 mNG-HAP2 promastigotes, an average of only 3.0% of cells expressed mNG-HAP2 in the untreated condition at day 1 post-inoculation, whereas at the same time point 23.4% of the L747 irradiated cells expressed mNG-HAP2 (*Figure 6C*). By day 2 post-inoculation, mNG-HAP2$^+$ cells in the irradiated cultures were detected at frequencies closer to those of untreated cultures (*Figure 6C*), indicating that the irradiated cells transiently upregulate HAP2 expression during log-phase growth. A similar pattern of increased frequency of mNG-HAP2$^+$ cells was observed for L747 parasites treated with either H$_2$O$_2$ (*Figure 6D*) or MMS (*Figure 6E*), although in each case the elevated expression persisted until day 4 of culture, and the effect was especially pronounced in the MMS-treated cells (60.2% mNG-HAP2$^+$). For the MA37 reporter line, HAP2 expression was again elevated on day 1 post-inoculation as cells transitioned into log-phase growth (*Figure 6H–J*). In this case, however, the proportions of mNG-HAP2$^+$ cells were not significantly different at day 1 comparing treated and irradiated cultures (32.8% vs. 28.9%, p=0.8857 Mann–Whitney test), which was consistent in all stress treatments. By day 2, the relative frequencies were altered, with levels declining in the untreated cells and increasing in the irradiated cells (*Figure 6H*). Similar patterns of mNG-HAP2 expression were observed in cells treated with either H$_2$O$_2$ or MMS (*Figure 6I and J*), with MMS inducing a more long-lasting effect. The relatively high frequency of mNG-HAP2$^+$ cells in the untreated MA37 cultures was surprising given the scRNA-seq comparisons showing the large expansion of HAP2-expressing cells in the irradiation-induced cluster, and suggests that HAP2 can be expressed under nonstress conditions even if not necessarily within the same population of promastigotes. By contrast, HAP2 expression in the L747 parent appears more dependent on induction by exposure to stress. The HAP2 expression profiles of the reporter constructs seem to more accurately reflect the pseudo-bulk RNA-seq

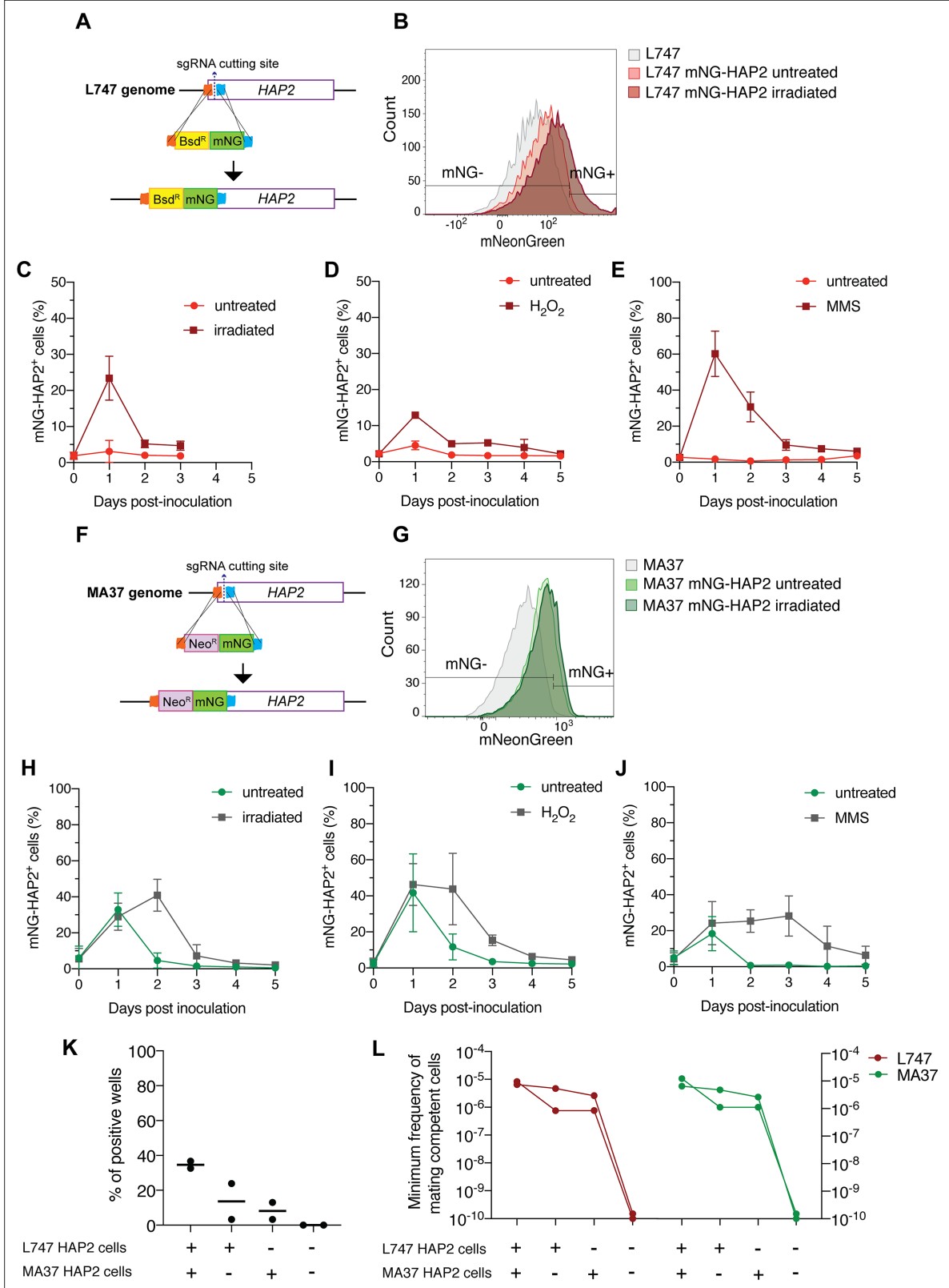

**Figure 6.** HAP2 protein expression in *L. tropica* promastigotes is associated with in vitro hybridization capacity. Schematic representation of the mNeonGreen (mNG)-HAP2 fusion reporter construct generated in the L747 strain (**A**) and in the MA37 strain (**F**). Stationary phase promastigotes of either L747 mNG-HAP2 or MA37 mNG-HAP2 parental lines were used to inoculate fresh cultures that were immediately divided into two flasks, with one flask remaining untreated and the other exposed to 6.5 Gy of γ-radiation or supplemented with 250 µM of H$_2$O$_2$ or 0.005% of methyl methanesulfonate

*Figure 6 continued*

(MMS). Flow cytometry histogram plots showing the fluorescence intensity of mNG-HAP2 in L747 (**B**) and in MA37 (**G**) 1 day post-inoculation, with or without irradiation. Proportion of mNG-HAP2$^+$ cells over time in L747 mNG-HAP2 cultures with or without exposure to γ-radiation (**C**), to H$_2$O$_2$ (**D**) or to MMS (**E**), n = 3 replicates. Proportion of mNG-HAP2$^+$ cells in MA37 transfectants after exposure to γ-radiation (**H**), to H$_2$O$_2$ (**I**), or to MMS (**J**), n = 4 replicates. (**K**) Proportion of culture wells exhibiting growth of LMA hybrids after in vitro crosses using the different indicated pairwise combinations of FACS-sorted mNG-HAP2$^+$ and mNG-HAP2$^-$ cells 1 day after irradiation; data are presented as mean values (black line) and individual measurements. (**L**) Minimum frequency of hybridization-competent cells for both L747 and MA37 parental strains in in vitro crosses using different combinations of FACS-sorted mNG-HAP2$^+$ and mNG-HAP2$^-$ cells (n = 2 independent experiments). Frequencies collected from each single experiment are linked by a line. Red lines and green lines represent data from L747 and MA37 parental strains, respectively.

The online version of this article includes the following source data and figure supplement(s) for figure 6:

**Source data 1.** Effect of different stress treatments on the percentage of mNG+ cells during growth of mNG-HAP2 *L. tropica* transfectants in vitro.

**Source data 2.** Percentage of hybrid-positive wells after mixing sorted mNG-HAP2+ and/or mNG-HAP2- *L. tropica* promastigotes.

**Figure supplement 1.** Representative flow cytometry analysis used to validate hybrids generated by crossing different combinations of mNG-HAP2$^+$ and mNG-HAP2$^-$ sorted cells from parental strains expressing either tdTomato or mCherry in the small ribosomal subunit rRNA locus (SSU).

comparisons (*Figure 5A*), which show a relatively small difference between irradiated and untreated MA37 cells, and a large difference between irradiated and untreated L747 cells.

We submitted the parental mNG-HAP2 reporter lines to γ-radiation and purified mNG-HAP2$^+$ and mNG-HAP2$^-$ cells by fluorescence-activated cell sorting (FACS) 1 day post-irradiation (*Figure 6B and G*), and co-cultured different combinations of the selected populations. Due to technical restrictions in the number of HAP2$^+$ cells that could be sorted, approximately 10-fold fewer promastigotes of each parent were used in these crosses compared to those performed using unsorted cells (*Supplementary file 1*). Verification of positive growing hybrids was performed using flow cytometry analysis to detect cells positive for both mCherry and tdTomato (*Figure 6—figure supplement 1*). Co-cultures that were established using the mNG-HAP2$^+$ cells from both parents produced the highest frequency of hybrids (34% positive wells), with a minimum frequency of hybridization-competent cells 7.5-fold the frequency detected in crosses using unsorted cells (*Figures 6K, L , and 1D*, *Supplementary file 1*). Interestingly, the presence of just one parental HAP2$^+$ cell population was sufficient to generate hybrids, albeit at lower frequencies (16 and 11% positive wells; 0.2–0.37-fold the minimum frequency of hybridization-competent cells vs. the double positive pair). Critically, co-cultures seeded with the mNG-HAP2$^-$ cells from both parents failed to generate any hybrids.

## Discussion

Reproduction in *Leishmania* includes a cryptic sexual cycle that operates among the extracellular promastigote stages developing in the sand fly vector. The mating-competent cells, and in particular the existence of a gametic stage, have yet to be identified. Studying the biology of mating in *Leishmania* would be greatly facilitated if these events could be reproduced using cells from axenic cultures, and we have recently achieved the recovery of stable hybrids generated between cultured promastigotes entirely in vitro (*Louradour et al., 2020*). While these initial findings established that mating-competent forms can arise in axenic culture, the mating frequencies were extremely low, and only a single pairwise combination of *L. tropica* parental lines would successfully hybridize. In the current studies, we demonstrate that under various conditions of environmental stress, including exposure to H$_2$O$_2$, MMS, or γ-radiation, the in vitro hybridization capacity of the *L. tropica* strains could be markedly enhanced and extended to other *L. tropica* strains and multiple *Leishmania* species, including *L. infantum, L. donovani,* and *L. braziliensis*. The products of these cell fusion events were in each case full-genome, polyploid hybrids. By single-cell RNA-seq analysis, we could identify transcriptionally unique populations of promastigotes whose numbers were drastically increased in each parental line following exposure to γ-radiation, and for which specific meiotic gene homologues, including the ancestral gamete fusogen HAP2, were found to be upregulated. By generating reporter constructs for HAP2, we could select for promastigotes that could hybridize or not in vitro.

Our findings add to the list of facultatively sexual eukaryotes, including yeast and green alga, that exhibit more sexual events under stress conditions. As each of the stress conditions that we imposed

are known to produce DNA damage, the results also support studies in varied model eukaryotes, suggesting that DNA repair is a central adaptive function of meiotic recombination (*Bernstein et al., 2018*; *Ram and Hadany, 2016*; *Schoustra et al., 2010*). Homologous chromosome pairing during meiosis provides an undamaged, template chromosome for DNA repair. The generation and repair of double-strand breaks is evidenced in our in vitro hybrids by the substantial number of chromosomes for which there were changes within the chromosome in the relative contribution of the parental alleles. A role for SPO11 protein in the induction of programmed DNA double-strand breaks is argued to be a universal feature of meiotic crossover events in eukaryotes (*Keeney et al., 1997*). However, meiotic recombination induced by X-irradiation in *S. cerevisiae* and in *C. elegans* was shown to occur even when SPO11 was absent or nonfunctional (*Dernburg et al., 1998*; *Thorne and Byers, 1993*). In our studies, expression of the homologous *SPO11* gene in *L. tropica* was not upregulated in the irradiated, hybridization-competent cells.

In order to identify hybridization-competent cells present in *Leishmania* cultures, we performed single-cell RNA sequencing of L747 and MA37 cultured cells exposed or not to γ-radiation. To our knowledge, this is the first application of single-cell transcriptomics in *Leishmania*, although it has been applied to other kinetoplastid pathogens to reveal the heterogeneity of the parasite populations arising in their mammalian hosts or insect vectors (reviewed in *Briggs et al., 2021*). Our analyses revealed transcriptomic heterogeneity within and between the parental *L. tropica* lines in the promastigote cultures. The heterogeneity likely reflects cells in different phases of the cell cycle, as well as differences in the developmental stage progression of the cultured promastigotes. The extensive differences in the transcriptomes of cells comparing the parental lines, apparent in both the PCA and UMAP visualizations, are surprising given the genetic similarity between these *L. tropica* strains (*Iantorno et al., 2017*) and likely reflect their different culture histories.

More meaningful to the current studies is the cluster heterogeneity comparing the untreated versus irradiated cells, which for each parental line revealed the emergence of a transcriptionally unique population in the irradiated cultures. Probing the clusters for enriched transcripts related to meiotic processes identified homologues of HAP2, GEX1, and RAD51, involved respectively in cell fusion, nuclear fusion, and DNA repair, that were largely confined to the unique clusters that emerged in the irradiated cultures. Even though a large fraction of the transcriptional shift is likely explained as part of a more global cellular mechanism to recover from stress caused by the irradiation, the expansion of cells expressing HAP2 transcripts in treated samples, the co-regulation of other meiotic genes (a HAP2-like FusM homologue, GEX1, and RAD51), and validation in vitro using HAP2 reporters, all strongly suggest that these cells are associated with the enhanced hybridization frequency observed post-treatment. The possible role in genetic exchange of other genes coordinately expressed with HAP2 but without any described meiotic function will be investigated in future studies. Of note, RNA-binding proteins, such as RBP5 and ZC3H11, were found to be upregulated in these cells and could have novel regulatory functions during *Leishmania* mating. Critically, we did not find significant upregulation of other meiosis-specific markers, including *SPO11, DMC1,* and *MND1.* This suggests that cell fusion might be a response to DNA stress that is independent of a meiotic program, especially as the meiotic reduction steps were for the most part absent in the stress-induced hybrids. Nevertheless, a similar finding was recently reported in an scRNA-seq analysis of *T. brucei* stages in the tsetse fly salivary glands, in which the gamete cell cluster was defined based solely on the upregulation of HOP1 and HAP2 (*Hutchinson et al., 2021*). It is possible that some of these markers of gametogenesis are expressed within a narrow time frame, and by analyzing single-cell gene expression at a single time point post-irradiation, we may have missed their expression peak in what is likely a rare cell population. In addition, expression of several genes may not have been detected due to 'dropout' events or excessive zeros, which has often been described in this type of approach due to a lower sequencing depth when compared to bulk RNA-seq (*Choi et al., 2020*).

HAP2 is a type I transmembrane protein that displays the same three-dimensional fold as class II viral fusion proteins. It plays a key role in the membrane fusion of gametes or mating cells that was first identified in plants, and is broadly conserved from protists to invertebrates (*Cole et al., 2014*; *Fédry et al., 2017*; *Johnson et al., 2004*; *Liu et al., 2008*; *Mori et al., 2006*). HAP2 expression was recently reported in *T. brucei*, both in gametes and in meiotic intermediates (*Peacock et al., 2021*). In the current work, we generated reporter constructs for HAP2 in L747 and MA37 and could show enhanced expression or altered expression kinetics in the stressed cultures. By sorting the cells into

mNG$^+$ and mNG$^-$ populations, we could confirm that HAP2 expression is a marker for hybridization-competent cells since hybridization failed unless the co-cultures included mNG$^+$ selected cells from one or both of the parents. To the extent that HAP2 is itself required for hybridization, then the apparent requirement for HAP2 expression on just one of the fusing cells is consistent with a number of studies showing that it can function unilaterally; either on the mating type minus gamete in *Chlamydomonas* algae (*Fédry et al., 2017*; *Liu et al., 2008*) or on the male gamete in *Plasmodium* (*Liu et al., 2008*), the ciliated protozoan *Tetrahymena thermophila* (*Cole et al., 2014*), or the plant *Arabidopsis thaliana* (*Johnson et al., 2004*; *Mori et al., 2006*). It is interesting that the MA37 parent showed a relatively high frequency of HAP2-expressing cells even absent exposure to the exogenous stress conditions. This may explain why this *L. tropica* strain hybridizes with such high efficiency in flies (*Inbar et al., 2019*), and why we are able to obtain some in vitro hybrids even without exogenous stress so long as MA37 is a fusion partner.

Contrary to the in vitro LMA hybrids that were previously generated, which were diploid, triploid, and tetraploid (*Louradour et al., 2020*), the irradiation-facilitated LMA hybrids, as well as the other intra- and interspecies hybrids described in this report, are in their great majority tetraploid. These results strongly suggest fusion between the two diploid parental cells, and that the stress conditions that promoted the fusogenic potential of the cells may have produced unusual features leading to unreduced gamete formation, which is a common mechanism underlying polyploidization in plants (*De Storme and Geelen, 2013*). Attempts to condition the tetraploid hybrids to complete a putative meiotic program by passing them through flies or by re-exposure to stress conditions failed to reduce their ploidy, showing that their polyploid state is relatively stable. As tetraploid progeny are occasional products of hybridization events in flies, it is also possible that tetraploid intermediates, as described for some fungi (*Heitman, 2010*; *Roman et al., 1955*), are a normal part of the mating process in *Leishmania,* with the ploidy reduction steps somehow lacking in the in vitro protocol. Since in the related protist *T. brucei,* haploid gametic cells have been identified prior to fusion in cells localized to the salivary gland of the tsetse fly vector (*Peacock et al., 2014*), we favor a comparable sexual process can operate during *Leishmania* development in sand flies. The fact that a few triploid hybrids, also observed in flies, were generated by the in vitro protocol could mean that a haploid gamete was produced by at least one of the parents and may encourage additional manipulations of the culture conditions to promote a more complete and uniform sexual cycle in vitro.

The absence of random segregation and genome-wide recombinations associated with a reductional meiotic program will undermine the utility of the hybrids for use in SNP association studies. It is interesting, nonetheless, that the hybrids seem to have retained an ability for homologue pairing and recombination after fusion, as evidenced by the breaks within chromosomes in the relative contributions of the parental alleles. Since the maintenance of the tetraploid state (two genome complements from each parent) would be expected to mask recombinations generated by reciprocal crossover events typically associated with meiosis I, the data suggest noncrossovers or gene conversion as a mechanism of DNA repair that would result in gene exchange in only one of the homologs. The asymmetric parental contributions observed both within and between different chromosomes could also have arisen by chromosome loss or gain at the level of individual somy, which is well described for *Leishmania* genomes during vegetative growth (*Dujardin et al., 2007*; *Sterkers et al., 2011*). Aneuploidy, including mosaic aneuploidy, is a constitutive feature of the *Leishmania* genome (*Sterkers et al., 2012*). Since gene dosage can directly control the levels of gene expression in *Leishmania* and impact phenotypes (*Iantorno et al., 2017*; *Prieto Barja et al., 2017*), the chromosome-specific patterns of aneuploidy and recombination in the hybrids can be used for linkage studies. It is notable that the progeny that were recovered are in almost every case distinct from the other progeny clones generated in the same experimental cross with respect to the combinations of the relative parental contributions to individual chromosomes that they possess. The distinct haplotypes observed in each of the 17 hybrids generated between the *L. donovani* strains from India and Sri Lanka are a case in point. As these parental strains produce, respectively, visceral and cutaneous forms of disease in humans and in mice (*Kariyawasam et al., 2018*; *Karunaweera et al., 2003*), it may be possible to map the genes controlling the behavior of the progeny clones in mice by genome-wide association studies. At a minimum, the admixture of these genomes should permit assessment as to whether or not the cutaneous or visceral tropisms are expressed as a dominant trait.

The stress-facilitated, in vitro hybridization protocol described in this report removes the requirement for sand flies as the main constraint to generating large numbers of genomic hybrids between *Leishmania* species and strains, and offers a transformative approach for genetic analysis to understand the extraordinary phenotypic diversity of the genus.

## Materials and methods

**Key resources table**

| Reagent type (species) or resource | Designation | Source or reference | Identifiers | Additional information |
|---|---|---|---|---|
| Gene (species) (*Leishmania tropica*) | Hapless2 (Hap2) | TritrypDB | LmjF.35.0460 | Also called GCS1 |
| Strain, strain background (*L. tropica*) | L747 RFP-Hyg | *Inbar et al., 2019* | MHOM/IL/02/LRC-L747 | https://doi.org/10.1371/journal.pgen.1008042 |
| Strain, strain background (*L. tropica*) | MA37 GFP-Neo | *Inbar et al., 2019* | MHOM/JO/94/MA37 | https://doi.org/10.1371/journal.pgen.1008042 |
| Strain, strain background (*Leishmania major*) | LV39 RFP-Hyg | *Akopyants et al., 2009* | MRHO/SU/59/P-strain | https://doi.org/10.1371/journal.pgen.1008042 |
| Strain, strain background (*L. major*) | FV1-Sat | *Akopyants et al., 2009* | MHOM/IL/80/Friedlin | https://doi.org/10.1371/journal.pgen.1008042 |
| Strain, strain background (*Leishmania infantum*) | LLM320 RFP-Hyg | *Romano et al., 2014* | MHOM/ES/92/LLM-320; isoenzyme typed MON-1 | https://doi.org/10.1073/pnas.1415109111 |
| Strain, strain background (*L. tropica*) | Moro RFP-Hyg | This paper | MHOM/AF/19/Moro | Patient from Afghanistan; cell line maintained in D. Sacks lab |
| Strain, strain background (*Leishmania donovani*) | Mongi RFP-Hyg | This paper | MHOM/IN/83/Mongi-142 | Patient from India; cell line maintained in D. Sacks lab |
| Strain, strain background (*L. donovani*) | SL2706 GFP-Neo | This paper | MHOM/LK/19/2706 | Patient from Sri Lanka; cell line maintained in D. Sacks lab |
| Strain, strain background (*Leishmania braziliensis*) | M1 GFP-Neo | This paper | MHOM/BR/00/BA779 | Patient from Brazil; cell line maintained in D. Sacks lab |
| Strain, strain background (*L. braziliensis*) | M1 RFP-Hyg | This paper | MHOM/BR/00/BA779 | Patient from Brazil; cell line maintained in D. Sacks lab |
| Strain, strain background (*L. braziliensis*) | RicX GFP-Neo | This paper | MHOM/PE/19/RicX | Patient from Peru; cell line maintained in D. Sacks lab |
| Strain, strain background (*L. braziliensis*) | RicX RFP-Hyg | This paper | MHOM/PE/19/RicX | Patient from Peru; cell line maintained in D. Sacks lab |
| Genetic reagent (*L. tropica*) | L747-mNG-HAP2 | This paper | MHOM/IL/02/LRC-L747 | Cell line maintained in D. Sacks lab |
| Genetic reagent (*L. tropica*) | MA37-mNG-HAP2 | This paper | MHOM/JO/94/MA37 | Cell line maintained in D. Sacks lab |
| Genetic reagent (*L. tropica*) | L747-mNG-HAP2 SSU-tdTom | This paper | MHOM/IL/02/LRC-L747 | Cell line maintained in D. Sacks lab |
| Genetic reagent (*L. tropica*) | MA37-mNG-HAP2 SSU-mCherry | This paper | MHOM/JO/94/MA37 | Cell line maintained in D. Sacks lab |

*Continued on next page*

*Continued*

| Reagent type (species) or resource | Designation | Source or reference | Identifiers | Additional information |
|---|---|---|---|---|
| Recombinant DNA reagent (plasmid) | pA2-GFP-Neo | *Chagas et al., 2014* | | https://doi.org/10.1371/journal.ppat.1003923 |
| Recombinant DNA reagent (plasmid) | pA2-RFP-Hyg | *Chagas et al., 2014* | | https://doi.org/10.1371/journal.ppat.1003923 |
| Recombinant DNA reagent (plasmid) | pSSU-tdTomato-Neo | Dr Deborah Smith, University of York, UK | | |
| Recombinant DNA reagent (plasmid) | pLEXSY-cherry-Sat2 | Jena Bioscience | Cat # EGE-236 | |
| Chemical compound, drug | CM199 | *Inbar et al., 2017* | | https://doi.org/10.1128/mBio.00029-17 |
| Chemical compound, drug | G418 (Geneticin) | Thermo Fisher | Cat # J63871.AB | 'Neo' antibiotic |
| Chemical compound, drug | Hygromycin B | Sigma-Aldrich | Cat # H3274 | 'Hyg' antibiotic |
| Chemical compound, drug | Nourseothricin | Sigma-Aldrich | Cat # 74667 | 'Sat' antibiotic |
| Chemical compound, drug | Blasticidin | Fisher Scientific | Cat # 10264913 | 'Blast' antibiotic |
| Chemical compound, drug | Hydrogen peroxide | Sigma-Aldrich | Cat # 216763 | $H_2O_2$ |
| Chemical compound, drug | Methyl methane sulfonate | Sigma-Aldrich | Cat # 129925 | MMS |
| Chemical compound, drug | Propidium iodide | Sigma-Aldrich | Cat # P4170 | |
| Chemical compound, drug | Hoechst 33342 | Thermo Fisher | Cat # 62249 | |
| Commercial assay or kit | AMAXA Nucleofactor 4D | Lonza | Cat # V4XP-3024 | |
| Commercial assay or kit | DNeasy Blood and Tissue Kit | QIAGEN | Cat # 69504 | |
| Commercial assay or kit | TruSeq Nano DNA Library Prep kit | Illumina | Cat # 20015965 | |
| Commercial assay or kit | Chromium Next GEM Single Cell 3' kit v3.1 | 10X Genomics | Cat # NC1690752 | |
| Commercial assay or kit | dsDNA HS Assay Kit | Invitrogen | Cat # Q32854 | |
| Software, algorithm | FACSDIVA | BD Biosciences | RRID:SCR_001456 | |
| Software, algorithm | FlowJo v.10.7 | Becton, Dickinson and Company | RRID:SCR_008520 | |
| Software, algorithm | PAINT | *Shaik et al., 2021* | | https://doi.org/10.3390/genes12020167 |
| Software, algorithm | Cell Ranger v.5.0 | 10X Genomics | RRID:SCR_017344 | |
| Software, algorithm | Seurat R package v.4.0.3 | *Hao et al., 2021* | RRID:SCR_016341 | |
| Software, algorithm | RStudio v.1.4.1717 | RStudio, PBC | RRID:SCR_000432 | |
| Software, algorithm | GraphPad Prism v.8.0 | GraphPad | RRID:SCR_002798 | |

## *Leishmania* cultures and transfection

Promastigotes were grown in axenic cultures at 26°C in complete medium 199 (CM199) (*Inbar et al., 2017*). The following parental *Leishmania* lines have been previously described (*Inbar et al., 2019*): *L. tropica* L747 RFP-Hyg (MHOM/IL/02/LRC-L747), *L. tropica* MA37 GFP-Neo (MHOM/JO/94/MA37),

*L. major* LV39 RFP-Hyg (MRHO/SU/59/P-strain), *L. major* FV1 Sat (MHOM/IL/80/Friedlin), *L. infantum* LLM320 RFP-Hyg (MHOM/ES/92/LLM-320; isoenzyme typed MON-1). The following additional parental lines were generated for the purposes of this study: *L. tropica* Moro RFP-Hyg (MHOM/AF/19/Moro), isolated from a cutaneous lesion in a patient from Afghanistan; *L. donovani* Mongi RFP-Hyg (MHOM/IN/83/Mongi-142), isolated from a patient with visceral leishmaniasis in India; *L. donovani* SL2706 GFP-Neo (MHOM/LK/19/2706), isolated from a cutaneous lesion in a patient from Sri Lanka; *L. braziliensis* M1 GFP-Neo or RFP-Hyg (MHOM/BR/00/BA779), isolated from a cutaneous lesion in a patient from Brazil; and *L. braziliensis* RicX GFP-Neo or RFP-Hyg (MHOM/PE/19/RicX), isolated from a cutaneous lesion in a patient from Peru.

For the parental lines generated in this study, *Swa*I-digested pA2-GFP-Neo or pA2-RFP-HYG plasmids (*Chagas et al., 2014*) were gel purified and the linear fragment flanked by the small subunit rRNA (SSU) homology regions was transfected into log-phase promastigotes using an AMAXA Nucleofector 4D (Lonza). Parasites ($8 \times 10^6$ cells) were harvested, washed in PBS, and resuspended in 20 µL supplemented P3 primary cell buffer (Lonza). DNA fragments (0.5–1 µg) and cells were mixed in a Nucleofector Strip, and transfection was performed immediately using program FI-115. Parasites were incubated at room temperature for 2 min in 100 µL CM199 and for 16 hr in 5 mL CM199 before selection of positive transfectants using appropriate drug pressure (see section 'In vitro generation of hybrids').

Inoculation of new parasite cultures was performed by dilution of a previous culture that had reached early stationary growth into fresh CM199 (100–500 µL into 5–10 mL CM199). For the stress-inducing conditions, a newly inoculated culture was immediately divided into two flasks, with one flask remaining untreated and the other exposed to 6.5 Gy of γ-radiation or supplemented with 250 µM of hydrogen peroxide ($H_2O_2$) or 0.005% of MMS.

## In vitro generation of hybrids

*Leishmania* hybrids were generated following the protocol previously developed (*Louradour et al., 2020*). Briefly, cultures of parental strains carrying resistance to a different antibiotic marker were initiated in parallel. 1 day post-inoculation, the concentration of each culture was estimated by counting under a hemocytometer and equal volumes of the parental cultures were mixed together and immediately distributed in 96-well plates (100 µL/well). The number of cells distributed per well varied according to the experiment and are summarized in *Supplementary file 1*. The co-cultures were transferred into a selective medium containing both antibiotics in 24-well plates on the next day (100 µL in 1 mL). The following antibiotics were used for double-drug-resistant hybrid selection: Geneticin (neomycin analog; Thermo Fisher; Neo 50 µg/mL), hygromycin B (Sigma-Aldrich; Hyg 25 µg/mL), nourseothricin (Fisher Scientific; Sat 200 µg/mL), and blasticidin S (Fisher Scientific; Bsd 20 µg/mL). The hybrid nature of the double-drug-resistant lines was by verified by their co-expression of fluorescence markers, analyzed by flow cytometry using a FACSCanto II system and FACSDiva software (BD Biosciences) (*Figure 1—figure supplement 2*). For hybrids generated from parents that did not carry different fluorescence markers, the presence of both parental antibiotic resistance markers was validated by PCR on DNA extracted from hybrid clones (DNeasy Blood and Tissue Kit, QIAGEN), targeting the antibiotic resistance genes (see *Supplementary file 5* for the sequence of PCR primers). For testing the effects of stress-inducing conditions on the generation of hybrids, parental cultures were initiated in parallel to untreated cultures and immediately exposed to 6.5 Gy of γ-radiation or supplemented with 250 µM of $H_2O_2$ or 0.005% of MMS. Four repetitions of each experiment were performed. For graphic representation on a logarithmic scale of the minimal frequencies of mating competent cells in each parental culture (*Figures 1 and 5*), a minimal value of 1e-10 was added to each frequency where no hybrids were obtained.

## DNA staining, flow cytometry, and confocal microscopy

To estimate the ploidy of *Leishmania* hybrids, promastigotes at logarithmic growth phase were stained in PI followed by flow cytometry analysis, as previously described (Louradour et al.). Briefly, cells were fixed in 0.4% paraformaldehyde for 1 min at room temperature, sedimented, resuspended in 100 µL PBS, and permeabilized in 1 mL methanol for 15 min on ice. After centrifugation, cells were incubated in 1 mL PBS for 10 min at room temperature. Next, DNA was stained using 500 µL of a PI-RNase mix (13 mg/mL each) at room temperature for 10 min. Stained cells were washed in cold PBS and analyzed

on a FACSCanto II (BD Biosciences) using FACSDIVA software. Flow cytometry data were analyzed using FlowJo software v.10.7 (Becton, Dickinson and Company).

For microscopic visualization of nuclei and kinteplasts by DNA staining, parasites were fixed in 4% paraformaldehyde, centrifuged, and resuspended in 0.5% glycine and incubated 10 min at room temperature. They were then pelleted again, washed in PBS, and incubated 30 min in 10 µg/mL Hoechst 33342 (Thermo Fisher). The fixed parasites were then mounted on polylysine-coated glass microscope slides, and images were acquired on a SP8 confocal microscope (Leica Microsystems).

## Whole-genome sequencing analysis

DNA was purified from log-phase promastigote cultures using the DNeasy Blood and Tissue Kit (QIAGEN) according to the manufacturer's instructions and submitted to Psomagen for next-generation sequencing (Rockville, MD). DNA libraries were generated using TruSeq Nano DNA Library Prep kit (Illumina), and the 100-bp-paired-end reads were sequenced on a HiSeq 2500 (Illumina). FastQ files were quality checked with FastQC software. Read trimming and filtering were performed with Trimmomatics (*Bolger et al., 2014*), resulting in >7.8M reads per library. Mean sequencing coverage in the dataset was 33.5, according to Qualimap v.2.2.1 (*Supplementary file 6*). Paired-end reads were aligned to the closest reference genomes available from TriTrypDB (tritrypdb.org; *L. tropica* L590 v.50, *L. donovani* BPK282A1 v.50, *L. infantum* JPCM5 v.50, *L. braziliensis* M2904 v.50) using BWA v.0.7.17 with default parameters. The PAINT software suite (*Shaik et al., 2021*) was used to find and extract the homozygous SNP marker differences between each parental strain pair. The alleles and their frequencies in the different hybrid progenies were found using the getParentAllelFrequencies utility in PAINT. Chromosome somies were determined by calculating the normalized read depth with the ConcatenatedPloidyMatrix in PAINT. In the case of polyploid hybrids, the somy values were divided by 2 and multiplied by the estimated ploidy from the DNA content analysis and the parental contribution profile. Somies were reported as heatmaps generated using the gplots and ggplot2 packages in R. Circos plots of the inferred parental contribution were generated using the Circos software (*Krzywinski et al., 2009*). Sequence reads obtained from the *Lt*MA37 and *Lt*Moro parental strains and MoMA hybrids that did not map to the reference nuclear genome sequence were used in maxicircle kDNA inheritance analysis. Unmapped reads were aligned to the *L. major* Friedlin maxicircle reference sequence (leish-esp.cbm.uam.es) using BWA v.0.7.17. Maxicircle sequence read depth and alternative allele frequencies were visualized in Integrative Genomics Viewer v.2.8.10 (*Robinson et al., 2011*).

## Single-cell RNA-seq library preparation and sequencing

Promastigotes at early-log phase were treated with 6.5 Gy of γ-radiation and compared with untreated cultures using scRNA-seq. Cells were harvested by centrifugation 1 day post-irradiation/inoculation, and samples were prepared according to Chromium Next GEM Single Cell 3′ Reagent Kits v3.1 (10X Genomics) as per the manufacturer's instructions. Briefly, $5 \times 10^3$ cells were loaded onto the 10X Chromium Controller microfluidics system and combined into microdroplets with a unique bead carrying oligonucleotides containing an Illumina adapter, a 10× cell barcode, a Unique Molecular Identifier (UMI) and a poly(dT) sequence. Co-partitioned cells were submitted to lysis, followed by reverse transcription of poly-adenylated mRNA, cDNA amplification, enzymatic fragmentation, and size selection. Final cDNA library quality control was assessed using a 4200 TapeStation system (Agilent) and Qubit fluorometer and dsDNA HS Assay Kit (Invitrogen). Libraries were submitted to Psomagen (Rockville, MD), pooled, and sequenced using two lanes of HiSeqX (Illumina; 150 bp paired-end; read length 28 * 8 * 91). Two biological replicates of each *L. tropica* culture were sequenced.

## scRNA-seq data preprocessing and quality control

Raw sequencing data were processed using Cell Ranger software v.5.0 (10X Genomics). Sequence reads were demultiplexed into FastQ files using Cell Ranger *mkfastq*. Reads were quality checked using FastQC, trimmed, and filtered using Fastp for average read quality of at least 20 and minimum length of 20 bp. Since Cell Ranger currently does not provide a built-in reference *Leishmania* transcriptome, a custom reference was generated using *L. major* Friedlin genome v.50 fasta and gff annotation files from TritrypDB. Reliable 3′UTR annotation was necessary for appropriate mapping of the scRNA-seq reads as the 3′tag-based approach used here only targets the 3′ end region of transcripts and eventual polyA stretches in the middle of genes. Therefore, absence of 3′UTR annotation

available for *L. tropica* strains made it necessary to use the mapped 3′UTR coordinates from genetically close *L. major* (Dillon et al.). Using custom scripts (the source code file is provided), the *L. major* transcript coordinates were included in the *L. major* gff annotation file, so that the original coding sequence (CDS) annotations were extended to the predicted 3′ ends of transcripts and renamed as 'exon' to be read by Cell Ranger *mkgtf* and *mkref*. *L. major* mitochondrial DNA sequence (kDNA maxi-circle) was added to the reference for eliminating the cell barcodes associated with dying/dead cells later. The resulting custom reference transcriptome was used to align filtered reads and quantify the number of different UMIs for each gene using Cell Ranger *count*. Data from irradiated and untreated samples from the same strain were aggregated together using Cell Ranger *aggr*. The fraction of reads containing a valid barcode assigned to a cell and confidently mapped to the transcriptome varied between 87 and 94% in MA37 and 85.7 and 86.8% in L747 libraries. The estimated number of cells detected in each replicate was between 3245 and 7485 with 20,401–49,580 mean reads per cell (*Supplementary file 7*). The generated UMI count matrices were loaded into RStudio v.1.4.1717 and further processed using the Seurat R package v.4.0.3 (*Hao et al., 2021*). Low-quality cells were removed according to the following parameters: (1) cells in which maxicircle genes represent >10% of total expression, (2) cells with expression of fewer than 200 different transcripts, or (3) more than 5000 different transcripts. The first two filters were used to account for dying/dead cells and empty beads and the third for multiplets in the same droplet.

## Cell clustering and differential expression analysis

Filtered data were normalized and variance stabilized by regularized negative binomial regression using the *sctransform* utility (*Hafemeister and Satija, 2019*) for cell clustering, dimensionality reduction, and UMAP visualization in Seurat. This package provides a fast approach to help decrease possible batch effects in the clustering. Maxicircle gene expression was regressed out of the sctransform analysis to avoid effects on the data clustering. For differential gene expression analyses reported in violin plots, dot plots, and heatmaps, the raw UMI counts were normalized by library read depth, log-transformed, centered, and scaled (Z-scored). The two replicates of untreated and irradiated samples from the same strain were integrated and analyzed as a single Seurat object. The 3000 most variable genes were identified using *FindVariableFeatures* utility and used to perform PCA. Based on empirical data, the top seven principal components were used to build a shared nearest-neighbor graph and modularity-based clustering using the Louvain algorithm with a resolution of 0.34 using *FindNeighbors* and *FindClusters*, which resulted in seven clusters for both *L. tropica* strains. UMAP visualization was calculated using these seven neighboring points. Transcript markers for each cluster were identified by comparing the average expression and the percent expressed of each gene in the cells within a cluster against the rest of the cells in the sample using *FindMarkers*. For the pseudo-bulk quality control analysis at the sample level of the scRNA-seq replicates shown in *Figure 3A*, the DESeq2 R package was used. Briefly, *DESeqDataSetFromMatrix* was used to create a DESeq object using the raw UMI counts, which were then normalized and log-transformed using *rlog* to account for differences in library depth and transcript composition. PCA plots were generated to visualize the first two principal components that best explain the variance in the data. Additional comparisons between untreated L747 and MA37 scRNA-seq data were performed by identifying match anchor cells with *FindIntegrationAnchors* and integrating the data from the two strains using *IntegrateData* in Seurat (*Figure 3—figure supplement 2*).

## HAP2mNG protein expression reporters

The L747-mNG-HAP2 and MA37-mNG-HAP2 reporter lines were generated by inserting resistance markers for blasticidin and neomycin, respectively, followed by an N-terminal mNeonGreen tag by CRISPR/Cas9 gene editing, following the strategy previously described (*Beneke et al., 2017*). The correct insertion of the tag was validated by two PCRs: the first using a forward primer located in the 5′UTR part of the Hap2 gene and a reverse primer located in the middle of the ORF; the second using the same forward primer and a reverse primer located at the end of the resistance marker inserted with the mNeonGreen (see *Supplementary file 5* for the complete list of primers used for the CRISPR/Cas9 strategy). The expression of the tagged protein was followed day by day using flow cytometry. At least three repetitions were performed for each treatment (6.5 Gy of irradiation, 150 µM of $H_2O_2$, and 0.005% of MMS), made simultaneously with untreated controls. The estimation

of the proportion of cells expressing the mNG-HAP2 was estimated using the FlowJo software v10.7 (Becton, Dickinson and Company) using identically treated L747 T7Cas9 and MA37 T7Cas9 cultures as negative controls for normalization.

### In vitro hybridization of mNG-HAP2-selected cells

For in vitro hybridization experiments using HAP2+-sorted cells, the mNG-HAP2 reporter lines were transfected with *Pmel-Pacl*-digested pSSU-tdTomato-Neo (kindly provided by Dr. Deborah Smith, University of York, UK) or Swal-digested pLEXSY-cherry-Sat2 (EGE-236, Jena Bioscience) following the same procedure described for GFP-Neo and RFP-Hyg above (see section '*Leishmania* cultures and transfection'). Irradiated cultures of L747 mNG-HAP2 and MA37 mNG-HAP2 were sorted 1 day post-inoculation and irradiation based on the expression of mNG using L747 T7Cas9 and MA37 T7Cas9 cultures as negative controls of fluorescence expression. After sorting, the mNG-HAP2+ and mNG-HAP2- cells from each parent were mixed in various combinations and the co-cultures were incubated overnight at 26°C in 96-well plates (100 μL/well). The co-culture wells were transferred individually in double-drug-selective medium (Bsd+ Neo) the following morning. Verification of positive growing hybrids was performed using a BD LSRFortessa system and FACSDiva software (BD Biosciences) to detect cells positive for both mCherry and tdTomato (*Figure 6—figure supplement 1*). Data were analyzed using FlowJo software v10.7 (Becton, Dickinson and Company).

### Transmission electron microscopy

Promastigotes at logarithmic growth phase ($10^7$ parasites) were harvested by centrifugation, washed in cold PBS, and resuspended in 1.5 mL of fixative solution (2.5% glutaraldehyde in 0.1 M sodium cacodylate buffer). Fixed cells were submitted to NIAID Electron Microscopy Unit (Hamilton, MT). All subsequent steps were carried out using a Pelco Biowave microwave at 250 W under vacuum. After rinsing in buffer, the cells were post-fixed with 1% reduced osmium tetroxide and treated with 1% tannic acid. The cells were stained with uranyl acid replacement stain, dehydrated with ethanol, infiltrated with Epon-Araldite resin, and polymerized overnight in a 60°C oven. Ultrathin sections were cut using a Leica UC6 ultramicrotome and imaged on Hitachi7800 TEM using an AMT camera.

### Statistical analyses

All statistical analyses for the whole-genome sequencing data and scRNA-seq were performed using specific R packages in RStudio v.1.4.1717. GraphPad Prism v.8.0 was used for statistical analysis of other data comparisons.

## Acknowledgements

We thank Margery Smelkinson and Owen Schartz (NIAID Biological Imaging Section) for the technical support with confocal imaging; Calvin Eigsti, Thomas Moyer, and Iyadh Douagi (NIAID Flow Cytometry Section) for the technical support with flow cytometry data acquisition and cell sorting and Vinod Nair (Electron Microscopy Unit, RML, NIH) for the support with electron microscopy image acquisition; Angela Cruz (FMRP, University of Sao Paulo) and Elise O'Connell (NIAID) for provision of the two *L. braziliensis* isolates, and Nilakshi Samaranayake and Hasna Riyal (Faculty of Medicine, University of Colombo) for coordinating the parasite propagation and passage of the Sri Lankan isolate. We also thank Najib El-Sayed (University of Maryland), Steve Beverley (Washington University), and P'ng Loke (NIAID, LPD) for helpful discussions. This work was supported in part by the Intramural Research Program of the National Institute of Allergy and Infectious Diseases, National Institutes of Health.

## Additional information

### Funding

| Funder | Grant reference number | Author |
|---|---|---|
| Division of Intramural Research, National Institute of Allergy and Infectious Diseases | | Andrea Paun |
| University of Colombo, Sri Lanka | NIAID Award Number U01AI136033 | Nadira Karunaweera |

The funders had no role in study design, data collection and interpretation, or the decision to submit the work for publication.

### Author contributions

Isabelle Louradour, Tiago Rodrigues Ferreira, Conceptualization, Data curation, Formal analysis, Investigation, Methodology, Writing – original draft, Writing – review and editing; Emma Duge, Formal analysis, Investigation, Methodology; Nadira Karunaweera, Conceptualization, Resources; Andrea Paun, Investigation; David Sacks, Conceptualization, Formal analysis, Funding acquisition, Supervision, Writing – original draft, Writing – review and editing

### Author ORCIDs

Tiago Rodrigues Ferreira ⓘ http://orcid.org/0000-0003-4894-1515
Nadira Karunaweera ⓘ http://orcid.org/0000-0003-3985-1817
David Sacks ⓘ http://orcid.org/0000-0002-7557-3124

### Decision letter and Author response

Decision letter https://doi.org/10.7554/eLife.73488.sa1
Author response https://doi.org/10.7554/eLife.73488.sa2

## Additional files

### Supplementary files

• Supplementary file 1. Calculations of the minimum frequencies of hybridization-competent cells for each L747 × MA37 cross.

• Supplementary file 2. Differentially expressed genes for each of the nine cell clusters identified in integrated untreated L747 and MA37 scRNA-seq data (after using *FindIntegrationAnchors* and *IntegrateData* commands) according to *FindAllMarkers* function in Seurat R package, with the following cutoff values: percentage of cells within a cluster expressing a given gene >10%, log2FC > 0.1, and Wilcoxon rank-sum test adjusted p-value < 0.05. Pct.1, percentage of cells in the cluster expressing a given gene; Pct.2, percentage of cells in all the other clusters expressing a given gene.

• Supplementary file 3. Differentially expressed genes for each of the seven cell clusters identified in L747 and MA37 (integrated irradiated and untreated samples) according to *FindAllMarkers* function in Seurat R package, with the following cutoff values: percentage of cells within a cluster expressing a given gene >10%, |log2FC| > 0.1, and Wilcoxon rank-sum test adjusted p-value < 0.05. Pct.1, percentage of cells in the cluster expressing a given gene; Pct.2, percentage of cells in all the other clusters expressing a given gene. Sheets containing the lists of the top 10 markers in each cluster or the genes upregulated in HAP2+ cells in clusters L-cluster2 and M-cluster1 vs. all the Hap2- cells in each strain are presented as additional tabs.

• Supplementary file 4. List of genes upregulated in both L747 cluster 2 and MA37 cluster 1 with the following threshold cutoffs: percentage of cells within a cluster expressing a given gene >10%, log2FC > 0.1, and Wilcoxon rank-sum test adjusted p-value < 0.05.

• Supplementary file 5. List of the primers used in this work.

• Supplementary file 6. Summary statistics of whole-genome sequencing read and alignment quality.

• Supplementary file 7. Summary statistics of single-cell RNA-sequencing reads and alignment quality.

• Transparent reporting form

• Source code 1. Code for preparing files for irradiated and untreated Ltrop scRNA-seq data analysis.

## Data availability

The raw sequence data containing reads from the 51 WGS samples and 8 scRNA-seq samples sequenced are deposited in the SRA database with Accession numbers PRJNA756557 and PRJNA756571, respectively. The source data files for Figures 1 and 6 are provided. Summary statistics on the sequencing data are available in Supplementary files 6 and 7.

The following dataset was generated:

| Author(s) | Year | Dataset title | Dataset URL | Database and Identifier |
|-----------|------|---------------|-------------|-------------------------|
| Sacks D | 2022 | raw sequence data containing reads from the 51 WGS samples | https://www.ncbi.nlm. nih.gov/bioproject/? term=PRJNA756557 | NCBI BioProject, PRJNA756557 |
| Sacks D | 2022 | scRNA-seq samples sequenced | https://www.ncbi.nlm. nih.gov/bioproject/? term=PRJNA756571 | NCBI BioProject, PRJNA756571 |

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
