## [Editor Report]

In this paper, the authors show that the ability of *Leishmania* promastigotes (a life-cycle stage found in sandflies) to fuse with each other is greatly enhanced after treatments that induce DNA damage. Although the fusion-competent cells express the gamete fusogen HAP2, the parasites do not undergo meiosis and the cells that result from fusion of two diploid organisms are mostly tetraploid; diploid progeny were not recovered. These observations are fundamentally interesting, and could be used for some types of investigations of gene function.

---

## [Decision Letter]

**Decision letter after peer review:**

Thank you for submitting your article "Stress conditions promote the mating competency of *Leishmania* promastigotes in vitro marked by expression of the ancestral gamete fusogen HAP2" for consideration by *eLife*. Your article has been reviewed by 4 peer reviewers, including Christine Clayton as the Reviewing Editor and Reviewer #1, and the evaluation has been overseen by Patricia Wittkopp as the Senior Editor. The following individual involved in review of your submission has agreed to reveal their identity: Marc Ouellette (Reviewer #2).

Essential revisions:

1) Please temper the claims. All referees agreed that this is really interesting but might not actually be "mating" in the usual sense, since there is no meiosis. This, combined with the inability to get diploids, also clearly compromise the usefulness of the resulting hybrids. So you also need to write very clearly what can, and cannot, be done with the method as it stands. (Just finding out whether characteristics are dominant or recessive? Any other possibilities?) Please change the title by replacing the word "mating" with "cell fusion" (or similar).

2) Otherwise all referees had various suggestions for improvement. Please make sure the huge Table is submitted as a dataset, the pdf conversion is a disaster, and fix the statistical presentation. No experiments required.

*Reviewer #1:*

In this paper the authors show that the ability of Leishmania promastigotes to fuse with each other is greatly enhanced after treatments that induce DNA damage. Although the fusion-competent cells express the gamete fusogen HAP2, the parasites do not undergo meiosis and the cells that result from fusion of two diploid organisms are mostly tetraploid. I actually wonder, therefore, whether this process can actually be called "mating" or whether it would be more correct to call it "cell fusion". It is not clear how useful this will be in practice since no meiosis appears to occur and diploid cells could not be recovered.

I personally think that the title should be modified, as it would be more accurate to use "cell fusion" rather than "mating".

In the discussion the authors should:

a) discuss the lack of other sexual/meisosis markers more critically. Could this be some sort of stress response rather than really triggering of the mating pathway?

b) Make more precise suggestions as to how this could be used. We no longer need mating for linkage studies since we have the genome sequences. We can transfect in modified genes and we can delete or mutate them. What, actually, would you use this for? Finding genes linked to any particular process by selection won't be possible unless the genomes can be reduced again. I guess you might find out that a phenotype was dominant but you wouldn't be able to assign it to a particular gene.

Abstract – The fact that hybrids were mostly tetraploid should be mentioned – it's more specific than "polyploid". In fact, specifically that the hybrids appear mostly to have retained both parental genomes.

Figures: Please make sure that all axis labels in all Figures can be read easily at 1x magnification. Also make all labels in one whole Figure the same size. People will be trying to read this on a laptop and will not want to have to magnify to 300% in order to understand the graphs. I found 6-point text – and less. For example, Figure 6 suffers from a very wide range of font sizes and formats. The numbers on B and G are tiny and in K and L the only important part is the (currently microscopic) superscripts. For the x-axis why not just put a large "+" or "-" and Use "L7547 HAP2" and "MA37 HAP2" as the left-hand labels? And for the y-axis in L you could put "log 10" and then use the superscripts as the main label.

Statistical analysis on Figures:

Generally please say how many replicates there were for each measurement in all Figures. If there are less than 5 replicates please show individual measurements (e.g. FIgure 6 H-K). I know it's been common practice for decades to use standard deviations when there are three measurements, but it's invalid and often does not really reflect what is observed. In Figure 6K the significance measurements must be removed, they are invalid.

Figure 1A, B, C: These should not be column graphs. You could use a box plot (though four measurements is rather low for that), or else just show the individual measurements. The distribution of points is clearly not normal so the standard deviations are invalid, please remove. Actually I'm not sure whether these are necessary at all since D,E and F are much more informative.

.

Figure 5A, B, G, H – this would be much easier to understand if the gene designations were used instead of the gene IDs. The IDs can be kept in the Legend only. Similarly in page 14 line 17 – also mention which genes were NOT up-regulated, since they might be a clue as to the lack of meiosis. How do the transcriptomes of mating-competent Leishmania compare with those seen from single-cell sequencing of T.. brucei? (Just released on line – see Hutchison et al., https://doi.org/10.1371/journal.ppat.1009904).

Figure 6: What is "inoculation" and what was the state of the cells before "inoculation"? Is it dilution of dense cells into new medium? Why do the authors think HAP2 expression increases when MA37 is "inoculated"? Were the cells mock-treated and could this be a source of stress? The stress treatments are missing from the methods section.

Supplementary Table 2 is an 85-page pdf. I wonder if this was an error, since as a pdf it is impossible to manipulate in any way. Please submit it in spreadsheet form and include gene annotations, and with the commas replaced by decimal points. The average log2 fold change would be easier to read if it were rounded to 4 decimal places. In the Table legend please say what the various clusters are. It would also be useful to highlight the "top ten markers" described on page 11 of the manuscript. A separate sheet could list the genes that are co-expressed with HAP2 – that would also be useful for people who are interested in looking for regulatory motifs.

"can be readily grown" – I'm old-fashioned, and find the split infinitive rather ugly. 'Can be grown readily" or "can readily be grown" are (to my eyes) preferable.

p3 Line 37 – "in vitro" should be italicised.

p4 Line 22 – how low? A number would be nice.

p4 Line 30 – how many cells per well?

p9 Line 13 – should be "a median number of 1,079 different mRNAs detected" (not "genes expressed").

p11 Is LmjF.31.1440 also a pseudogene in other species? (is it a multigene family?).

p11 line 47 "being expressed in" should be "expression being detected in".

*Reviewer #2:*

Louradour et al., have followed up on their exciting Leishmania sex studies. In a previous study they have shown that hybrids could be obtained in vitro, but only when working with L. tropica. Now they show that this hybrid formation is increased significantly in cells subjected to treatments known to damage DNA and that importantly it can be extended to other Leishmania species. Furthermore, and elegantly, they have shown that these hybrids are found only in cells expressing HAP2.

I am not asking for further experiments (a lot of competent work is included in this study) but a number of points would merit clarifications. Indeed, it remains to be seen whether the tetraploid state of the hybrids will allow SNP discovery linked to phenotypes. I would presume that inkage studies would require massive recombination for reducing the complexity of the hybrids. I am curious to know whether the authors have attempted to cross a tetraploid hybrid with a wild-type cell (I realize that this will require the insertion of new markers) but this back-cross type of experiment would go a long way to support the hypothesis that in vitro sex could be helpful for GWAS studies. Can the author also comment on why hybrids cannot be obtained with L. major? Were any L. major hybrids (in sand-fly or natural isolates) described previously?

*Reviewer #3:*

Leishmania parasites have been shown to produce genetic hybrids after co-transmission via the sand fly vector of genetically different parental strains of the same or different species. in vitro production of hybrids has also been shown for one species, L. tropica. Here the authors set out to investigate whether the production of hybrids in vitro could be enhanced by stress induction via irradiation or chemical means.

Pairwise mixtures of drug resistant Leishmania strains were analysed for production of double drug resistant hybrids before or after stress induction. The number of hybrids greatly increased following stress induction in both L. tropica and other Leishmania species tested. Analysis of the DNA contents of hybrids showed the vast majority to be 4N compared to 2N parental lines, with few 3N and no 2N hybrids; this contrasts with sand fly co-transmission experiments, where the majority of hybrids are 2N with occasional polyploids. Genome sequencing and SNP analysis indicated that both sets of parental chromosomes were present in the 4N hybrids and they appear to be fusions of the parental cells, albeit with some evidence of chromosomal recombination. The authors further investigated the stress response by single cell RNA sequencing of L. tropica cultures, and showed that a cluster of cells characterised by expression of sex-related genes such as HAP2 and GEX1, increased in number. Tagging HAP2 with a fluorescent marker demonstrated that the number of HAP2 expressing cells increased after cell cultures were irradiated and cultures enriched for HAP2-expressing cells showed increased production of hybrids. The authors' interpretation of their data is that stress induces formation of sex cells in vitro with increased production of hybrids in mixed cultures. There are indeed more hybrids after stress induction, but the vast majority are whole genome fusions of the two parental lines. The number of cells expressing HAP2 is also increased after stress induction. However, neither of these phenomena may be part of the natural sexual reproduction process in Leishmania.

The manuscript is well written overall and the figures are clear.

Might be informative to include kinetoplast maxicircle data for hybrids to determine whether they have kDNA from both parents as well as nuclear genomes.

The single cell RNA sequencing data show an interesting difference between L. tropica strains in producing metacyclics in vitro, which is presently buried in Supp Figure 6 – perhaps more could be made of this?

*Reviewer #4:*

Louradour et al. present exciting data on an approach to greatly improve the efficiency of purely in vitro experimental hybridisation in Leishmania, which might help male forward genetics approaches practical in this important parasite group. As well as this methodological step forward, the data presented add to our understanding of the basic reproductive biology of Leishmania. The conclusions of this paper are well-supported by the data and analysis presented, and the writing and presentation is of a very high quality throughout.

One thing that makes the paper slightly difficult to follow at one place is the separate clustering of single-cell data from the two different hybrid lines. It wIs there no way to either cluster genes from the two strains in a single analysis, or somehow identify which clusters represent the same cell groups across the strain. These clusters could then be numbered the same across the lines, which would clarify the writing. That might then also allow us to get some handle on to what extent similar clusters show overlapping gene expression patterns, which is lost in the current analysis. I appreciate that the number of significantly DE genes differs greatly between the two lines, but it would be good to drill a little down into whether this is due to substantially different patterns of expression between the two lines, or differences in the power to detect differential expression.

One missing piece that slightly dilutes the impact of this work is that it is not yet completely clear that what is happening in the in vitro hybrids is identical to the clear crossing-over in meiosis seen in experimental fly infections and that is evident in natural populations. The authors observe a few apparent intra-chromosomal changes in parental coverage, there appear to be rather few (a handful per line) : rather less than we expect to see in normal meioses, and perhaps so few as to be consistent with some kind of mitotic crossing-over. This slight uncertainty on whether the in vitro system really reflects the situation in flies also slightly dents the relevance of the finding that hybrids here are highly polyploid fusions of parental cells.

I guess that many crossing-overs are not associated with changes in parental coverage, but it seems a shame that no attempt has been made to directly infer cross-overs in these data. I presume the heterozygosity is too low, and this would be challenging in polyploid-ish cells in any case. To my mind this rather reduces the utility of in vitro hybridisation and thus the impact of this work.

Finally, nothing is really said about the 'other clusters' in the scRNA-seq data. While it is nice to see scRNA-seq being used to test a specific hypothesis rather than purely descriptively, it seems a slightly unfortunate omission to now say something about these data more generally, given that I think it is the first scRNA-seq in Leishmania. I suspect the authors could – and perhaps should – argue that including this would dilute the main message of this manuscript and be beyond the scope of what they are trying to achieve here.

I have some detailed comments on the text that might help the authors revise the manuscript.

p2 line 11 – is it 'mating competency' or 'mating competence'.

p3 line 6 – 'to infection' – this clause isn't quite clear to me. I think it would be better with 'and infection'?

p3 line 29 – 'have also been found in Leishmania ancestral lineages' makes it sound like they are not found in modern Leishmania, which i don't think is the intention.

p3 line 44 – 'is proposed' -> 'is proposed to be'.

p4 line 42 – is this calculation still really valid in the 'high yield' experiments?

p5 lines 47-49. Is this enough information about the parasite strains involved?

p6, line 18 – 'promastigotes -> 'promastigote'.

line 20 – 'The diploid parental lines', is it known that the parental lines are diploid?

line 31:These results suggest that tetraploid intermediates, as have been described in some fungi, might reflect a normal part of the sexual cycle in Leishmania. Is this true?

p8, line 16 – 'parental somies are in each case close to 2n ' – isn't this by assumption (p6, line20)? Isn't the observation more that parents vary little in somy between chromosomes.

p8, line 24 – the 'gain of somy' is particularly interesting here, as it doesn't quite fit the model of 'fusion followed by somy loss'.

p8, line 44 – 'inferring ' I think should read 'implying'.

p8 line 41-44. Does asymetric inheritance of a partial chromosome be explained *just* by recombination, or does it also involve some variation in transmission. Is this explicable in an F1?

p12, lines 21-22. 'irradiation-induced M-cluster1 and L-cluster2 (HAP2+ cells vs the total HAP2- cell population; |log2FC|>0.25, adj. p<0.05). ' – not completely clear which cluster (or both?) the enrichment reported relates to.

p14, line 16 – it’s not completely clear to me what 'common to these clusters ' means here – is it just expressed in both clusters M-cluster1 and L-cluster2 , or only expressed there? I think the former, but it might be worth a few words to clarify.

p14, line 44-47. 'The HAP2 expression profiles of the reporter constructs seem to more accurately reflect the pseudo-bulk RNA-seq comparisons (Figure 5A), which show a relatively small difference between irradiated and untreated MA37 cells, and a large difference between irradiated and untreated L747 cells.' – is this contrast between lines reflected in the proportions of MA37 and L747 cells in the relevant scRNA-seq clusters?

p17, line 16 – missing a closing parenthesis.

p21, line 34-36. Missing some details of how the UTRs were mapped over.

p22, lines 1-4. Missing some details of low-quality cells. was the proportion of maxicircle reads bimodal? How many cells were removed in these filters?

p24, line 1 – I think it is Najib, not Nagib?

[Editors' note: further revisions were suggested prior to acceptance, as described below.]

Thank you for submitting your article "Stress conditions promote *Leishmania* hybridization in vitro marked by expression of the ancestral gamete fusogen HAP2 as revealed by single-cell RNAseq" for consideration by *eLife*. Your article has been reviewed by 3 peer reviewers, and the evaluation has been overseen by a Reviewing Editor and Patricia Wittkopp as the Senior Editor. The following individuals involved in review of your submission have agreed to reveal their identity: Marc Ouellette (Reviewer #2).

The reviewers were happy with the revisions, but one requested that more neutral terms (e.g. "cell fusion" instead of "mating") be used consistently throughout.

Please follow the suggestions of Reviewer 3 with regard to neutral language throughout the text.

*Reviewer #2:*

The response to the reviewer was adequate and the authors modified the manuscript accordingly.

*Reviewer #3:*

In the revised version, the title now better reflects the gist of the paper, but this is not completely followed through in the text as mating and sexual processes are often used to describe what is observed in vitro, e.g. line 22 abstract, line 83 to promote Leishmania mating, line 86 capacity for in vitro mating, line 89 mating-competent cells, line 430 mating events, line 438 exhibit more sexual events under stress conditions, line 478 enhanced mating frequency, line 505 marker for mating competent cells, concluding sentence lines 565-68. I would have preferred a more neutral account and a critical discussion of whether the observed hybridisation is part of the natural sexual reproduction process in Leishmania, analogous to the mating observed in sand flies. A model for how the observed tetraploid intermediates are incorporated into the normal Leishmania sexual cycle (line 167) would be helpful.

---

## [Author Response]

Reviewer #1:In this paper the authors show that the ability of Leishmania promastigotes to fuse with each other is greatly enhanced after treatments that induce DNA damage. Although the fusion-competent cells express the gamete fusogen HAP2, the parasites do not undergo meiosis and the cells that result from fusion of two diploid organisms are mostly tetraploid. I actually wonder, therefore, whether this process can actually be called "mating" or whether it would be more correct to call it "cell fusion". It is not clear how useful this will be in practice since no meiosis appears to occur and diploid cells could not be recovered.

We think the point about mating vs fusion is fair, and we have removed ‘mating’ from the title while emphasizing the genome hybridization aspects of the cell fusion events. We agree that the utility of the tetraploid hybrids in terms of SNP association studies is certainly compromised. We argue, nonetheless, that there are sufficient differences in the genotypes of different progeny clones due to their distinct patterns of aneuploidy and homologous recombination that linkage studies may be possible, especially given the ease with which so many different hybrids can now be generated.

I personally think that the title should be modified, as it would be more accurate to use "cell fusion" rather than "mating".

We have changed the title to emphasize the genome hybridization aspects of the cell fusion events. i.e. the admixture of heterologous genomes.

In the discussion the authors should:a) discuss the lack of other sexual/meisosis markers more critically. Could this be some sort of stress response rather than really triggering of the mating pathway?

We have expanded on this important point the discussion, lines 482-494:

“Critically, we did not find significant up-regulation of other meiosis-specific markers, including *SPO11*, *DMC1* and *MND1*. […] In addition, expression of several genes may not have been detected due to “dropout” events or excessive zeros, which has often been described in this type of approach due to a lower sequencing depth when compared to bulk RNA-seq (Choi et al., 2020).”

b) Make more precise suggestions as to how this could be used. We no longer need mating for linkage studies since we have the genome sequences. We can trasnfect in modified genes and we can delete or mutate them. What, actually, would you use this for? Finding genes linked to any particular process by selection won't be possible unless the genomes can be reduced again. I guess you might find out that a phenotype was dominant but you wouldn't be able to assign it to a particular gene.

As efficient as reverse genetic tools are in *Leishmania*, they have so far failed to identify genes controlling such fundamental traits as tissue tropism, pathogenicity and virulence. These traits are likely under polygenic control, and will be difficult for targeted mutagenesis to address. An unbiased, forward genetics approach offers an important strategy, although the reviewer is correct that in the absence of meiotic reduction and recombination, SNP association studies will be far more challenging. We observed, nonetheless, that following fusion of the parental genomes, there are extensive changes in the relative contribution of the parental chromosomes due to homologous recombinations and copy number variation at the level of individual somy. We argue that because of the genotype differences between progeny clones, plus the relative ease with which large numbers of hybrids can now be generated, that linkage studies may be possible. And as the reviewer suggests, assessing the dominance of a particular trait should also be possible. All of these points are raised in the discussion, lines 538-563.

Abstract – The fact that hybrids were mostly tetraploid should be mentioned – it's more specific than "polyploid". In fact, specifically that the hybrids appear mostly to have retained both parental genomes.

We have specified in the abstract that the hybrids were mostly tetraploid.

Figures: Please make sure that all axis labels in all Figures can be read easily at 1x magnification. Also make all labels in one whole Figure the same size. People will be trying to read this on a laptop and will not want to have to magnify to 300% in order to understand the graphs. I found 6-point text – and less. For example, Figure 6 suffers from a very wide range of font sizes and formats. The numbers on B and G are tiny and in K and L the only important part is the (currently microscopic) superscripts. For the x-axis why not just put a large "+" or "-" and Use "L7547 HAP2" and "MA37 HAP2" as the left-hand labels? And for the y-axis in L you could put "log 10" and then use the superscripts as the main label.

Each of the figures has been modified as suggested.

Statistical analysis on Figures:Generally please say how many replicates there were for each measurement in all Figures. If there are less than 5 replicates please show individual measurements (e.g. Figure 6 H-K). I know it's been common practice for decades to use standard deviations when there are three measurements, but it's invalid and often does not really reflect what is observed. In Figure 6K the significance measurements must be removed, they are invalid.

Done.

Figure 1A, B, C: These should not be column graphs. You could use a box plot (though four measurements is rather low for that), or else just show the individual measurements. The distribution of points is clearly not normal so the standard deviations are invalid, please remove. Actually I'm not sure whether these are necessary at all since D,E and F are much more informative.Figure 5A, B, G, H – This would be much easier to understand if the gene designations were used instead of the gene IDs. The IDs can be kept in the Legend only. Similarly in page 14 line 17 – also mention which genes were NOT up-regulated, since they might be a clue as to the lack of meiosis.

The meiosis gene homologues that were not upregulated are discussed lines 482-494.

How do the transcriptomes of mating-competent Leishmania compare with those seen from single-cell sequencing of T.. brucei? (Just released on line – see Hutchison et al., https://doi.org/10.1371/journal.ppat.1009904).

Comparison with the Hutchison paper is discussed lines 485-488.

Figure 6: What is "inoculation" and what was the state of the cells before "inoculation"? Is it dilution of dense cells into new medium? Why do the authors think HAP2 expression increases when MA37 is "inoculated"? Were the cells mock-treated and could this be a source of stress? The stress treatments are missing from the methods section.

Inoculation refers to the initiation of the fresh cultures with the treated or untreated cells. The cells used for inoculation were from an early stationary culture. This information is more clearly provided in the Material and Methods and the legend to figure 6. The untreated cells were not “mock-treated”. HAP2 expression appears to be transient, upregulated during log phase growth and down regulated in stationary phase cells. This point is better communicated on lines 372-378 of the results.

Supplementary Table 2 is an 85-page pdf. I wonder if this was an error, since as a pdf it is impossible to manipulate in any way. Please submit it in spreadsheet form and include gene annotations, and with the commas replaced by decimal points. The average log2 fold change woould be easier to read if it were rounded to 4 decimal places.

Done, Sup File 3.

In the Table legend please say what the various clusters are.

Apart from two clusters that can be defined by enrichment of procyclic or metacyclic specific markers, we are not able to provide meaningful descriptions of the other clusters.

It would also be useful to highlight the "top ten markers" described on page 11 of the manuscript. A separate sheet could list the genes that are co-expressed with HAP2 – that would also be useful for people who are interested in looking for regulatory motifs.

Done, Sup File 3, separate tab.

"can be readily grown" – I'm old-fashioned, and find the split infinitive rather ugly. 'Can be grown readily" or "can readily be grown" are (to my eyes) preferable.p3 Line 37 – "in vitro" should be italicised.p4 Line 22 – how low? A number would be nice.

Done.

p4 Line 30 – how many cells per well?

Specified in Sup. File 1.

p9 Line 13 – should be "a median number of 1,079 different mRNAs detected" (not "genes expressed").

Done.

p11 Is LmjF.31.1440 also a pseudogene in other species? (is it a multigene family?).

We have provided additional information about this gene, lines 327-329.

p11 line 47 "being expressed in" should be "expression being detected in".

Done.

Reviewer #2:Louradour et al., have followed up on their exciting Leishmania sex studies. In a previous study they have shown that hybrids could be obtained in vitro, but only when working with L. tropica. Now they show that this hybrid formation is increased significantly in cells subjected to treatments known to damage DNA and that importantly it can be extended to other Leishmania species. Furthermore, and elegantly, they have shown that these hybrids are found only in cells expressing HAP2.I am not asking for further experiments (a lot of competent work is included in this study) but a number of points would merit clarifications. Indeed, it remains to be seen whether the tetraploid state of the hybrids will allow SNP discovery linked to phenotypes. I would presume that inkage studies would require massive recombination for reducing the complexity of the hybrids. I am curious to know whether the authors have attempted to cross a tetraploid hybrid with a wild-type cell (I realize that this will require the insertion of new markers) but this back-cross type of experiment would go a long way to support the hypothesis that in vitro sex could be helpful for GWAS studies. Can the author also comment on why hybrids cannot be obtained with L. major? Were any L. major hybrids (in sand-fly or natural isolates) described previously?

Regarding the utility of the hybrids for SNP association studies, please refer to our response to the public evaluation summary and to lines 538-563 of the discussion.

We have not attempted a cross with a tetraploid hybrid as yet.

Why our *L. major* strains will not hybridize in vitro remains an unanswered but intriguing question, especially as they do hybridize in sand flies, demonstrating that they possess the machinery for genetic exchange (Akopyants et al., Science, 2009; Inbar et al., PloS Gen, 2013).

Reviewer #3:Leishmania parasites have been shown to produce genetic hybrids after co-transmission via the sand fly vector of genetically different parental strains of the same or different species. in vitro production of hybrids has also been shown for one species, L. tropica. Here the authors set out to investigate whether the production of hybrids in vitro could be enhanced by stress induction via irradiation or chemical means.Pairwise mixtures of drug resistant Leishmania strains were analysed for production of double drug resistant hybrids before or after stress induction. The number of hybrids greatly increased following stress induction in both L. tropica and other Leishmania species tested. Analysis of the DNA contents of hybrids showed the vast majority to be 4N compared to 2N parental lines, with few 3N and no 2N hybrids; this contrasts with sand fly co-transmission experiments, where the majority of hybrids are 2N with occasional polyploids. Genome sequencing and SNP analysis indicated that both sets of parental chromosomes were present in the 4N hybrids and they appear to be fusions of the parental cells, albeit with some evidence of chromosomal recombination. The authors further investigated the stress response by single cell RNA sequencing of L. tropica cultures, and showed that a cluster of cells characterised by expression of sex-related genes such as HAP2 and GEX1, increased in number. Tagging HAP2 with a fluorescent marker demonstrated that the number of HAP2 expressing cells increased after cell cultures were irradiated and cultures enriched for HAP2-expressing cells showed increased production of hybrids. The authors' interpretation of their data is that stress induces formation of sex cells in vitro with increased production of hybrids in mixed cultures. There are indeed more hybrids after stress induction, but the vast majority are whole genome fusions of the two parental lines. The number of cells expressing HAP2 is also increased after stress induction. However, neither of these phenomena may be part of the natural sexual reproduction process in Leishmania.

The possibility that the stress-facilitated hybridization reflects cell fusion events unrelated to natural mating is an important point that has been raised by the other reviewers, and is discussed in the paragraph beginning on line 518 of the discussion.

The manuscript is well written overall and the figures are clear.Might be informative to include kinetoplast maxicircle data for hybrids to determine whether they have kDNA from both parents as well as nuclear genomes.

We agree that the maxicircle kDNA data would be informative to include. It can now be found in Figure 2—figure supplement 4 and referred to lines 214-219 of the results. It is interesting that maxicircle kDNA inheritance appears uniparental, consistent with our experimental crosses in flies and the vast majority of natural crosses described to date. In this respect the in vitro hybrids are not simply products of the fusion of diploid cells but share an important character of natural hybrids.

The single cell RNA sequencing data show an interesting difference between L. tropica strains in producing metacyclics in vitro, which is presently buried in Supp Figure 6 – perhaps more could be made of this?

The single cell RNA sequencing data was generated from cells obtained from log phase growth. Few metacyclic forms would be expected in these cultures and the sample comparisons were not designed to identify clusters representing different development stages of promastigotes. This point is clarified on lines 266-273 of the results. While the strain differences in the small cluster of cells representing metacyclics is interesting, we would prefer not to comment further on this given the constraints of our sampling bias.

Reviewer #4:Louradour et al. present exciting data on an approach to greatly improve the efficiency of purely in vitro experimental hybridisation in Leishmania, which might help male forward genetics approaches practical in this important parasite group. As well as this methodological step forward, the data presented add to our understanding of the basic reproductive biology of Leishmania. The conclusions of this paper are well-supported by the data and analysis presented, and the writing and presentation is of a very high quality throughout.One thing that makes the paper slightly difficult to follow at one place is the separate clustering of single-cell data from the two different hybrid lines. It wIs there no way to either cluster genes from the two strains in a single analysis, or somehow identify which clusters represent the same cell groups across the strain. These clusters could then be numbered the same across the lines, which would clarify the writing. That might then also allow us to get some handle on to what extent similar clusters show overlapping gene expression patterns, which is lost in the current analysis. I appreciate that the number of significantly DE genes differs greatly between the two lines, but it would be good to drill a little down into whether this is due to substantially different patterns of expression between the two lines, or differences in the power to detect differential expression.

The decision to perform the comparisons between untreated vs irradiated cells separately for each strain was due to the very limited representation of shared cell clusters between the untreated L747 and MA37 strains. The clustering of the integrated, untreated L747 and MA37 is now shown in Figure 3—figure supplement2A and referred to in the text in lines 275-286, emphasizing how distinctive their transcriptional profiles are at the single cell level. Therefore, to avoid additional unnecessary bias in the analysis of irradiation-induced changes, cells from each strain were clustered separately. The complete list of cluster markers for the analysis of integrated, untreated strains is now included in the new Sup. File 2.

One missing piece that slightly dilutes the impact of this work is that it is not yet completely clear that what is happening in the in vitro hybrids is identical to the clear crossing-over in meiosis seen in experimental fly infections and that is evident in natural populations. The authors observe a few apparent intra-chromosomal changes in parental coverage, there appear to be rather few (a handful per line) : rather less than we expect to see in normal meioses, and perhaps so few as to be consistent with some kind of mitotic crossing-over. This slight uncertainty on whether the in vitro system really reflects the situation in flies also slightly dents the relevance of the finding that hybrids here are highly polyploid fusions of parental cells.

We agree that the number of observable recombinations is less than expected from normal meiosis, and as we followed only the homozygous SNPs that are different between the parents, the recombinations will have occurred following fusion of the parental genomes, likely during subsequent mitotic divisions.

I guess that many crossing-overs are not associated with changes in parental coverage, but it seems a shame that no attempt has been made to directly infer cross-overs in these data. I presume the heterozygosity is too low, and this would be challenging in polyploid-ish cells in any case. To my mind this rather reduces the utility of in vitro hybridisation and thus the impact of this work.

We agree that the extra ploidy and low heterozygosity will make it difficult to appreciate many of the cross-overs that may have occurred, and that this will compromise the utility of the hybrids. Nonetheless, the cross-overs that were detectable, in addition to the differences in parental contributions at the level of individual chromosomes (both are now highlighted in the new circos plots shown in Figure 2 -Figure sup. 3), may still afford opportunities for genome wide association studies. This point is referred to in the text lines 208-212 of the results, and in the discussion in the paragraph beginning on line 538.

Finally, nothing is really said about the 'other clusters' in the scRNA-seq data. While it is nice to see scRNA-seq being used to test a specific hypothesis rather than purely descriptively, it seems a slightly unfortunate omission to now say something about these data more generally, given that I think it is the first scRNA-seq in Leishmania. I suspect the authors could – and perhaps should – argue that including this would dilute the main message of this manuscript and be beyond the scope of what they are trying to achieve here.

It is true that we struggled a bit to know how to balance the findings from the scRNA seq data that the comparisons were specifically designed to address, and the additional information that fell out of the first application of this method to *Leishmania*. Based on the reviewer’s comment, we have decided to expand our presentation and discussion of the scRNA seq data from the untreated cells to consider in more detail the heterogeneity revealed both within and between the two strains. See the paragraph beginning on line 253 of the results.

I have some detailed comments on the text that might help the authors revise the manuscript.p2 line 11 – is it 'mating competency' or 'mating competence'.

According to google scholar, ‘competence’ is more commonly used, so we have changed our usage in the text.

p3 line 6 – 'to infection' – this clause isn't quite clear to me. I think it would be better with 'and infection'?

Done.

p3 line 29 – 'have also been found in Leishmania ancestral lineages' makes it sound like they are not found in modern Leishmania, which i don't think is the intention.

We modified the text to read “…. found in trypanosomatid ancestral lineages and extant *Leishmania* species….”

p3 line 44 – 'is proposed' -> 'is proposed to be'.

Done.

p4 line 42 – is this calculation still really valid in the 'high yield' experiments?

We are not clear as to why the high yield experiments would impact the calculation of minimal mating frequency.

p5 lines 47-49. Is this enough information about the parasite strains involved?

More information regarding the strains is provided in Table 1 and in the Material and Methods.

p6, line 18 – 'promastigotes -> 'promastigote'.

Done.

line 20 – 'The diploid parental lines', is it known that the parental lines are diploid?

Yes, each of the parental lines is known to be close to 2n based on their propidium iodide staining profiles using a diploid reference strain (LmFn) for comparison.

line 31:These results suggest that tetraploid intermediates, as have been described in some fungi, might reflect a normal part of the sexual cycle in Leishmania. Is this true?

Yes, it is true that some fungi have a tetraploid meiotic cycle. And while we believe it is possible that *Leishmania* can use a similar process, it is not the normal process that we favor. See the paragraph beginning on line 518 of the discussion.

p8, line 16 – 'parental somies are in each case close to 2n ' – isn't this by assumption (p6, line20)? Isn't the observation more that parents vary little in somy between chromosomes.

Yes, by assumption with respect to the sequence analysis, and by inference based on the DNA content analysis.

p8, line 24 – the 'gain of somy' is particularly interesting here, as it doesn't quite fit the model of 'fusion followed by somy loss'.

Gain of somy, which we have also observed in our hybrids generated in sand flies, may presumably arise by chromosome non-disjunction during meiotic or mitotic divisions.

p8, line 44 – 'inferring ' I think should read 'implying'.

The sentence now reads ‘implying’.

p8 line 41-44. Does asymetric inheritance of a partial chromosome be explained just by recombination, or does it also involve some variation in transmission. Is this explicable in an F1?

We have argued that gene conversion involving chromosome homologues post-fusion is a mechanism to explain the observable differences in parental contributions of only part of a chromosome. Apart from partial chromosome loss in one parent prior to fusion, which seems unlikely, it is difficult to understand how variation in transmission would account for the asymetric inheritance observed.

p12, lines 21-22. 'irradiation-induced M-cluster1 and L-cluster2 (HAP2+ cells vs the total HAP2- cell population; |log2FC|>0.25, adj. p<0.05). ' – not completely clear which cluster (or both?) the enrichment reported relates to.

The text has been clarified to read “…we investigated the list of genes co-expressed with *HAP2* in each one of the irradiation-induced clusters, M-cluster1 and L-cluster2 (gene expression in *HAP2^+^* cells in each of these clusters vs the total *HAP2*^-^ cell population in each strain; |log2FC|>0.25, adj. *p*<0.05).

p14, line 16 – it’s not completely clear to me what 'common to these clusters ' means here – is it just expressed in both clusters M-cluster1 and L-cluster2 , or only expressed there? I think the former, but it might be worth a few words to clarify.

Yes, the former. The text has been clarified to read “Several genes that were upregulated in cells from L-cluster2 and M-cluster1 as a whole (Figure 4C)…”

p14, line 44-47. 'The HAP2 expression profiles of the reporter constructs seem to more accurately reflect the pseudo-bulk RNA-seq comparisons (Figure 5A), which show a relatively small difference between irradiated and untreated MA37 cells, and a large difference between irradiated and untreated L747 cells.' – is this contrast between lines reflected in the proportions of MA37 and L747 cells in the relevant scRNA-seq clusters?

Yes, in Figure 3D the proportion of irradiated L747 cells in cluster 2 is 47.5%, while the proportion of MA37 cells in cluster 1 is 36.2%. This information had been added to the text, line 295.

p17, line 16 – missing a closing parenthesis.

Done.

p21, line 34-36. Missing some details of how the UTRs were mapped over.

The text has been corrected to read “….the *L. major* transcript coordinates were included in the *L. major* gff annotation file, so that the original coding sequence (CDS) annotations were extended to the predicted 3’ ends of transcripts….”

p22, lines 1-4. Missing some details of low-quality cells. was the proportion of maxicircle reads bimodal? How many cells were removed in these filters?

The detailed information for each sample regarding the low quality cells and the number of cells removed in the filters are provided in Sup File 6. The proportion of maxicircle reads was not bimodal.

p24, line 1- I think it is Najib, not Nagib?

Done.

[Editors' note: further revisions were suggested prior to acceptance, as described below.]

The reviewers were happy with the revisions, but one requested that more neutral terms (e.g. "cell fusion" instead of "mating") be used consistently throughout.Please follow the suggestions of Reviewer 3 with regard to neutral language throughout the text.Reviewer #3:In the revised version, the title now better reflects the gist of the paper, but this is not completely followed through in the text as mating and sexual processes are often used to describe what is observed in vitro, e.g. line 22 abstract, line 83 to promote Leishmania mating, line 86 capacity for in vitro mating, line 89 mating-competent cells, line 430 mating events, line 438 exhibit more sexual events under stress conditions, line 478 enhanced mating frequency, line 505 marker for mating competent cells, concluding sentence lines 565-68. I would have preferred a more neutral account and a critical discussion of whether the observed hybridisation is part of the natural sexual reproduction process in Leishmania, analogous to the mating observed in sand flies. A model for how the observed tetraploid intermediates are incorporated into the normal Leishmania sexual cycle (line 167) would be helpful.

All of the references to mating that pertain to the hybrids generated in this report are now described as either hybridization or cell fusion.

A critical discussion of whether the observed hybridization is part of the natural sexual reproduction process in *Leishmania* is provided on page 13 of the discussion.